# The Life Cycle of *Aurelia aurita* Depends on the Presence of a Microbiome in Polyps Prior to Onset of Strobilation

Nadin Jensen,[a] Nancy Weiland-Bräuer,[a] Shindhuja Joel,[a] Cynthia Maria Chibani,[a] Ruth Anne Schmitz[a]

[a]Institute of General Microbiology, Christian-Albrechts University Kiel, Kiel, Germany

**ABSTRACT** *Aurelia aurita*'s intricate life cycle alternates between benthic polyp and pelagic medusa stages. The strobilation process, a critical asexual reproduction mechanism in this jellyfish, is severely compromised in the absence of the natural polyp microbiome, with limited production and release of ephyrae. Yet, the recolonization of sterile polyps with a native polyp microbiome can correct this defect. Here, we investigated the precise timing necessary for recolonization as well as the host-associated molecular processes involved. We deciphered that a natural microbiota had to be present in polyps prior to the onset of strobilation to ensure normal asexual reproduction and a successful polyp-to-medusa transition. Providing the native microbiota to sterile polyps after the onset of strobilation failed to restore the normal strobilation process. The absence of a microbiome was associated with decreased transcription of developmental and strobilation genes as monitored by reverse transcription-quantitative PCR. Transcription of these genes was exclusively observed for native polyps and sterile polyps that were recolonized before the initiation of strobilation. We further propose that direct cell contact between the host and its associated bacteria is required for the normal production of offspring. Overall, our findings indicate that the presence of a native microbiome at the polyp stage prior to the onset of strobilation is essential to ensure a normal polyp-to-medusa transition.

**IMPORTANCE** All multicellular organisms are associated with microorganisms that play fundamental roles in the health and fitness of the host. Notably, the native microbiome of the Cnidarian *Aurelia aurita* is crucial for the asexual reproduction by strobilation. Sterile polyps display malformed strobilae and a halt of ephyrae release, which is restored by recolonizing sterile polyps with a native microbiota. Despite that, little is known about the microbial impact on the strobilation process's timing and molecular consequences. The present study shows that *A. aurita*'s life cycle depends on the presence of the native microbiome at the polyp stage prior to the onset of strobilation to ensure the polyp-to-medusa transition. Moreover, sterile individuals correlate with reduced transcription levels of developmental and strobilation genes, evidencing the microbiome's impact on strobilation on the molecular level. Transcription of strobilation genes was exclusively detected in native polyps and those recolonized before initiating strobilation, suggesting microbiota-dependent gene regulation.

**KEYWORDS** *Aurelia aurita*, native microbiota, asexual reproduction, strobilation, recolonization, life cycle, reproduction

Address correspondence to Ruth Anne Schmitz, rschmitz@ifam.uni-kiel.de.

The authors declare no conflict of interest.

An animal host together with its associated natural microbiota forms a unit that is defined as a metaorganism (1, 2). The complex functional interplay between a host and its microbiota has a crucial impact on the physiology and proper development of the host (3–5), as well as on the host metabolism (6–8), the immune system (9), morphogenesis (10), reproduction (11–14), and environmental adaption (15–18). Metaorganismal studies allow for the unraveling of mechanisms by which a host-associated microbial community affects these processes. Changes in the abundance and diversity or the complete absence

of a specific microbiota can be associated with autoimmune diseases (19), obesity (20), metabolic disorders (21–23), and cancer (24, 25). However, the constitution and functionality of a host's-associated microbiome depends on environmental conditions (26, 27), such as the life stage (28) and the physiology of the host (1). The research on invertebrate-microbiota interactions, in particular, broadens the concept of the metaorganism, since it enables to disentanglement of fundamental mechanisms of host-microbe interactions as well as their underlying regulatory principles (2). Recently, we established the moon jellyfish *Aurelia aurita*, one of the most extensively studied Scyphozoan (29, 30), as a model for metaorganism research (11, 28, 31). The advantage of jellyfishes as model hosts lies in their simple body structure, which consists of only two tissue layers and an acellular layer: the endoderm, the ectoderm, and the mesoglea separating them (32). The surfaces of both epithelia are subject to microbial colonization and interaction. Despite its simple body structure, the jellyfish is distinguished by a complex life cycle. *A. aurita* undergoes complex metamorphic changes by alternating between a sessile polyp and a free-swimming medusa life stage. The transition from polyp to medusa is typically triggered by a seasonal temperature shift in spring and is characterized by the formation of constrictions starting on the apical part of the polyp; this developmental stage is known as early strobila. The elongated polyp undergoes complete segmentation within days and marks the late strobila phase. Individual segments are then released as single ephyrae (33–36). In a previous study, we identified distinct bacterial community patterns for *A. aurita*'s individual life stages, pointing to the significance of microorganisms within the polyp-to-medusa transition (28). At the molecular level, hormonal and neuronal signals act as nuclear hormone receptors, activating the metamorphosis (33, 37). It has been demonstrated that several proteins, including transcriptional regulators, participate in the induction of strobilation (33, 38–42). These include the retinoic X receptor (RxR), whose differential regulation correlates with strobilation (33, 38, 40). The transcription-regulating proteins *de novo* methyl transferase (DNMT) and patched (Ptc) are also implicated in the induction of strobilation (33, 39, 41, 42). Moreover, the bone morphogenetic protein (BMP) is engaged in the differentiation processes of the lobes and the gastrovascular system, whereas Wnt1 is a morphogen for the localization involved in the anterior-posterior axis formation in metazoans (43–45). The molecular machinery of *A. aurita* metamorphosis further includes the differential expression of the strobilation-specific genes *CL112*, *CL355*, *CL390*, and *CL631* (39, 42). The expression of these four genes is strongly upregulated in polyps that are about to enter the strobilation phase. *CL112* encodes a protein that contains an epidermal growth factor-like domain and is expressed in endodermal cells of each segment of the early strobila (39). The expression of *CL112* was detected in induced polyps even when they still lacked morphological evidence of strobilation. Hence, a function in inducing strobilation was suggested (39). The same was true for the gene *CL390* (43). Both *CL390* and *CL631* are expressed in ectodermal cells; they encode proteins with a signal peptide and several arginine repeats. *CL631* expression, however, was demonstrated at the apical border of the second segment in the ectoderm and possibly correlates with morphogens (33, 39, 42). Finally, *CL355* was shown to be expressed in the ectoderm at the posterior side of each segment (39, 42). Thus, *CL112*, *CL355*, *CL390*, and *CL631* seem to play a role in starting strobilation in *A. aurita*.

We recently described how the fitness of *A. aurita* is severely impaired in the absence of a native microbiota, including the halt of asexual reproduction (11). Recolonizing sterile polyps with a native polyp-derived microbiota could restore asexual development and reproduction. However, whether the developmental restoration of a sterile polyp by recolonization is possible at several developmental stages (polyp, early strobila, or late strobila) or whether the precise timing of the host-microbiota interaction is important remained to be determined. Moreover, the underlying molecular pathways were unknown.

Here, we aimed to identify the developmental stage(s) for the recolonization of sterile animals in order to restore asexual reproduction and to gain insights into the role of the microbiota in this process. Sterile animals were recolonized with the native microbiota at three different life stages: polyp prior to the initiation of strobilation, early strobila, and late

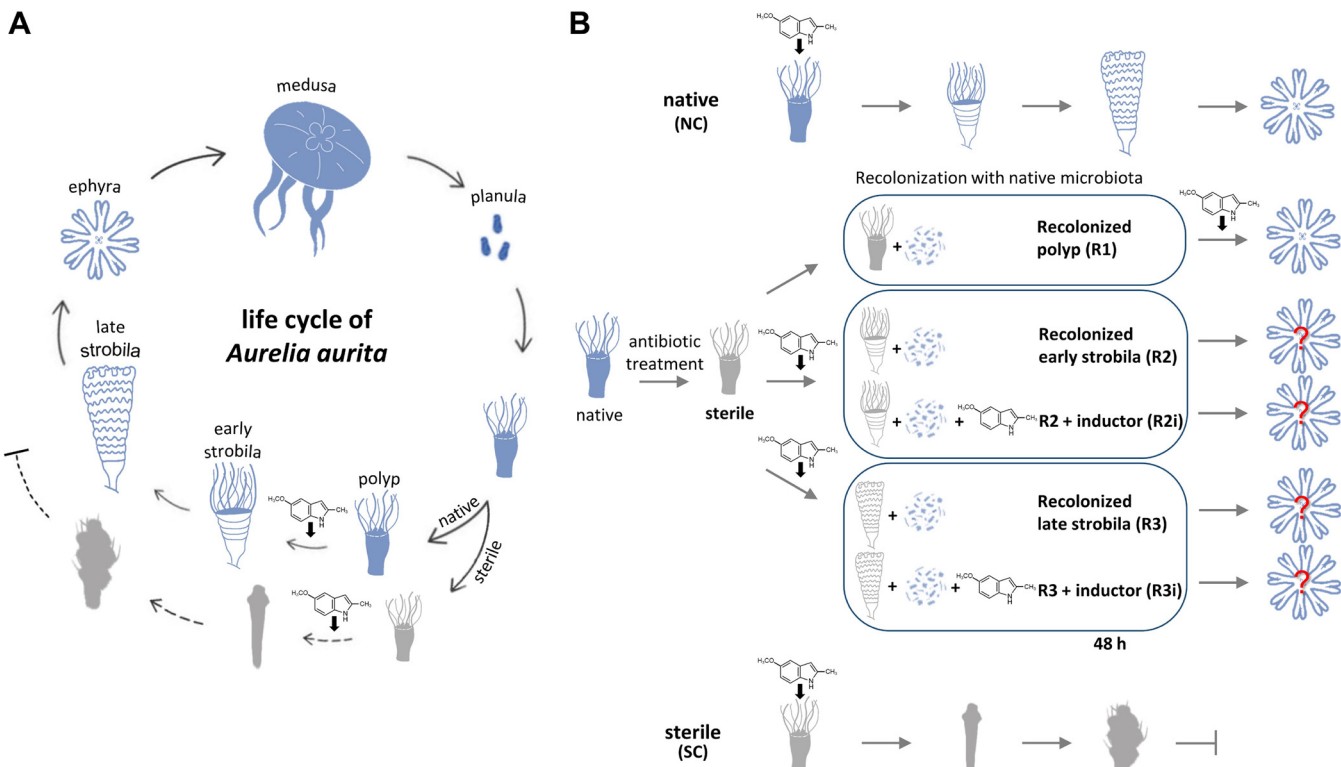

**FIG 1** Life cycle of *Aurelia aurita* and study design for recolonization at various developmental stages. (A) Summary of the life cycle of the moon jellyfish *Aurelia aurita*. Sexual reproduction of the matured medusa results in the release of planula larvae that develop into benthic polyps. Asexual reproduction includes the transition into early and then late strobila, forming bodily segments to release precursor medusa in the form of ephyrae. This offspring generation is impaired in the absence of a microbiota. Sterile animals showed malformations characterized by a pale color, slim body shape, and a lack of tentacles represented in a gray color of early and late strobila (11). (B) Experimental design to identify a permissible time point(s) of recolonization. Native polyps are induced with the chemical inducer for the onset of strobilation (native conditions [NC]), which was removed by washing after 72 h. Similarly, sterile polyps were induced, resulting in impaired strobilation (sterile conditions [SC]). Sterile polyps were recolonized prior to induction (R1). Developed sterile strobila stages (light gray images) were recolonized after washing off the inducer (R2, R3); in a second approach, the inducer was again added when recolonization was performed (constant presence of inducer; R2i, R3i).

strobila. The formation of strobilae and the release of ephyrae were evaluated, and phenotypic effects, like segment formation and color of strobilae and size, color, and shape of ephyrae, were recorded. The gene expression of developmental and strobilation genes was examined by reverse transcription-quantitative PCR (qRT-PCR) to determine the effect of the exact timing of recolonization for asexual reproduction, focusing on the molecular level.

## RESULTS

The life cycle of *A. aurita* is summarized in Fig. 1A. Sexual reproduction of the mature medusa results in the release of planula larvae that develop into benthic polyps. Asexual reproduction can be induced chemically to start the transition into early and then late strobila, forming bodily segments to release precursor medusa in the form of ephyrae. This offspring generation is impaired in the absence of a microbiota by a pale color, slim body shape, and a lack of tentacles but can be restored by recolonizing sterile polyps (11). The experimental setup to study the effect of the different timing of this recolonization is summarized in Fig. 1B. As a first step, sterile polyps were recolonized (treatment R1), and the associated microbiota was characterized.

**Recolonization of sterile polyps with native microbiota demonstrates host impact on microbial community structure.** The microbiota used as inoculum to recolonize sterile polyps was isolated from homogenized and filtered native polyps and characterized by amplicon sequencing of the V1-V2 region of the 16S rRNA gene. Following the recolonization of sterile polyps, their microbiota was characterized again and compared to that of native polyps (Fig. 2). In all three microbiota samples, the classes of Alphaproteobacteria and Gammaproteobacteria dominated. In the native polyps,

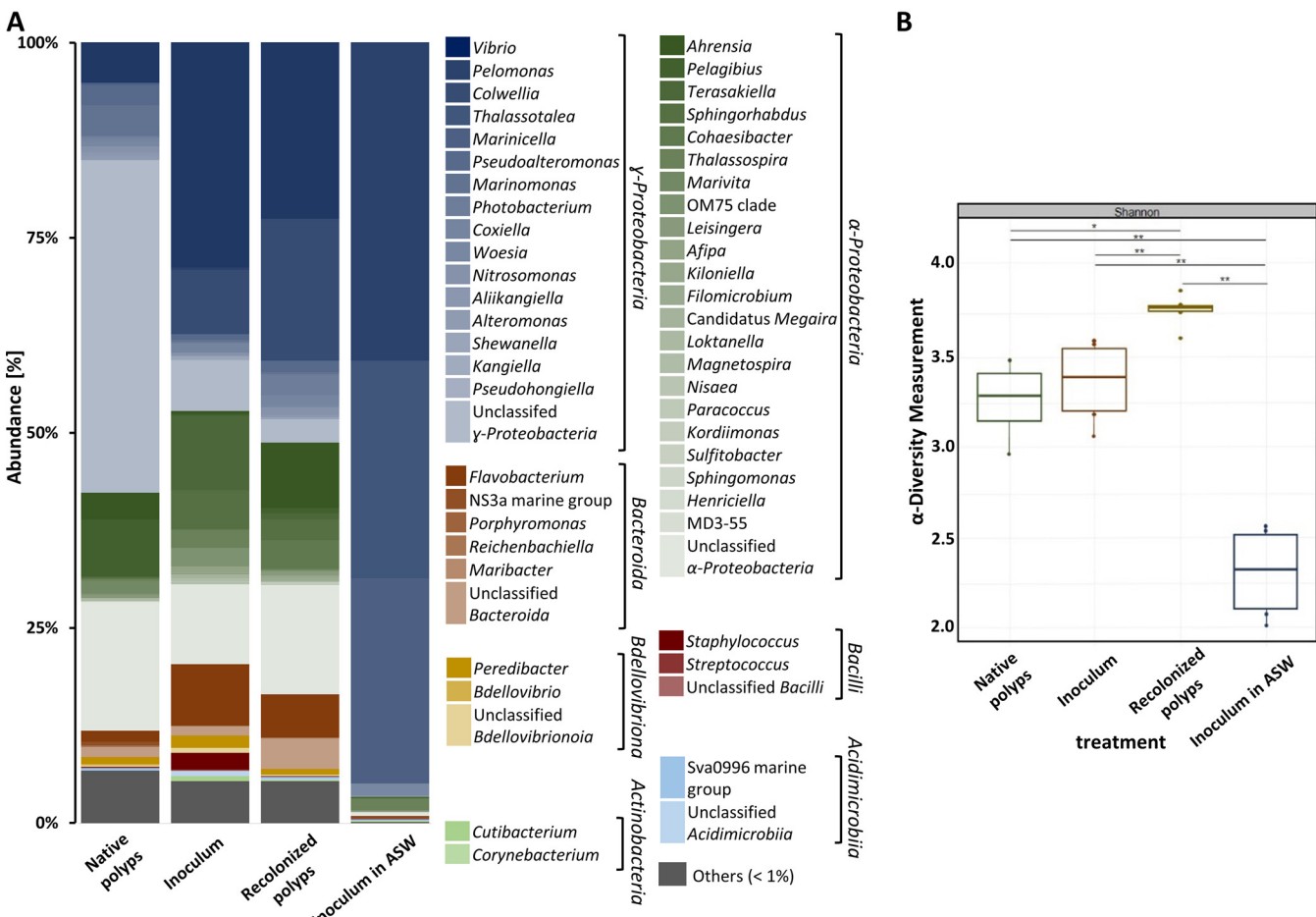

**FIG 2** Analysis of the microbial community of *Aurelia aurita*. (A) ASV abundance based on the V1-V2 region of the 16S rRNA gene of the bacterial communities derived from native polyps, the inoculum used for recolonization, obtained microbiota from recolonized sterile polyps, and a control of the inoculum incubated in ASW are presented at the genus level and normalized by the total number of reads per sample. Bar plots represent averages of six replicates per treatment. The threshold for the represented taxa is >1% relative abundance; consequently, taxa with a mean relative abundance of <1% across all samples are collectively reported as "Others." In the electronic version, a scroll over the colored box and a right mouse click will show the genus. (B) Box plot of α-diversity (Shannon index). Significance of comparisons is shown: *, $P < 0.05$; **, $P < 0.01$.

unclassified representatives of Gammaproteobacteria were dominant, followed by the genera *Pelagibus* and *Vibrio* (Fig. 2A, first column). The major representatives in the inoculum were (in descending order) *Vibrio*, *Terasakiella*, *Colwellia*, *Flavobacterium*, and *Sphingorhabdus*. The diversity of Alphaproteobacteria and Gammaproteobacteria was similar in native polyps and inoculum, resulting in no significant differences in the α-diversity (Fig. 2B). Thus, although differences in abundances were noted, the inoculum was considered similar to the microbiota of native polyps. Recolonizing sterile polyps with the inoculum shifted the bacterial community to an increased relative abundance of *Colwellia* and *Ahrensia*. Compared to the native polyps, recolonized polyps had larger fractions of *Vibrio*, *Ahrensia*, *Cohaesibacter*, and *Flavobacterium* (Fig. 2A). The α-diversity was significantly increased in recolonized polyps compared to native ones but showed less variation. Taxa present in native polyps but missing in the inoculum and recolonized polyps were exclusively grouped in the "other" category due to <1% abundance (Fig. 2B). Overall, the microbiota of recolonized polyps resembled that of native polyps. In contrast, the abundance of bacteria maintained in artificial seawater (ASW) declined in the absence of animals. The large difference in the community structure of the latter was reflected by a significantly lower α-diversity (Fig. 2B). In summary, recolonization of the sterile polyp with an inoculum derived from native microbiota resulted in a microbial community pattern similar to those observed in native polyps. Consequently, we predict similar phenotypic and molecular consequences for the polyp after recolonization.

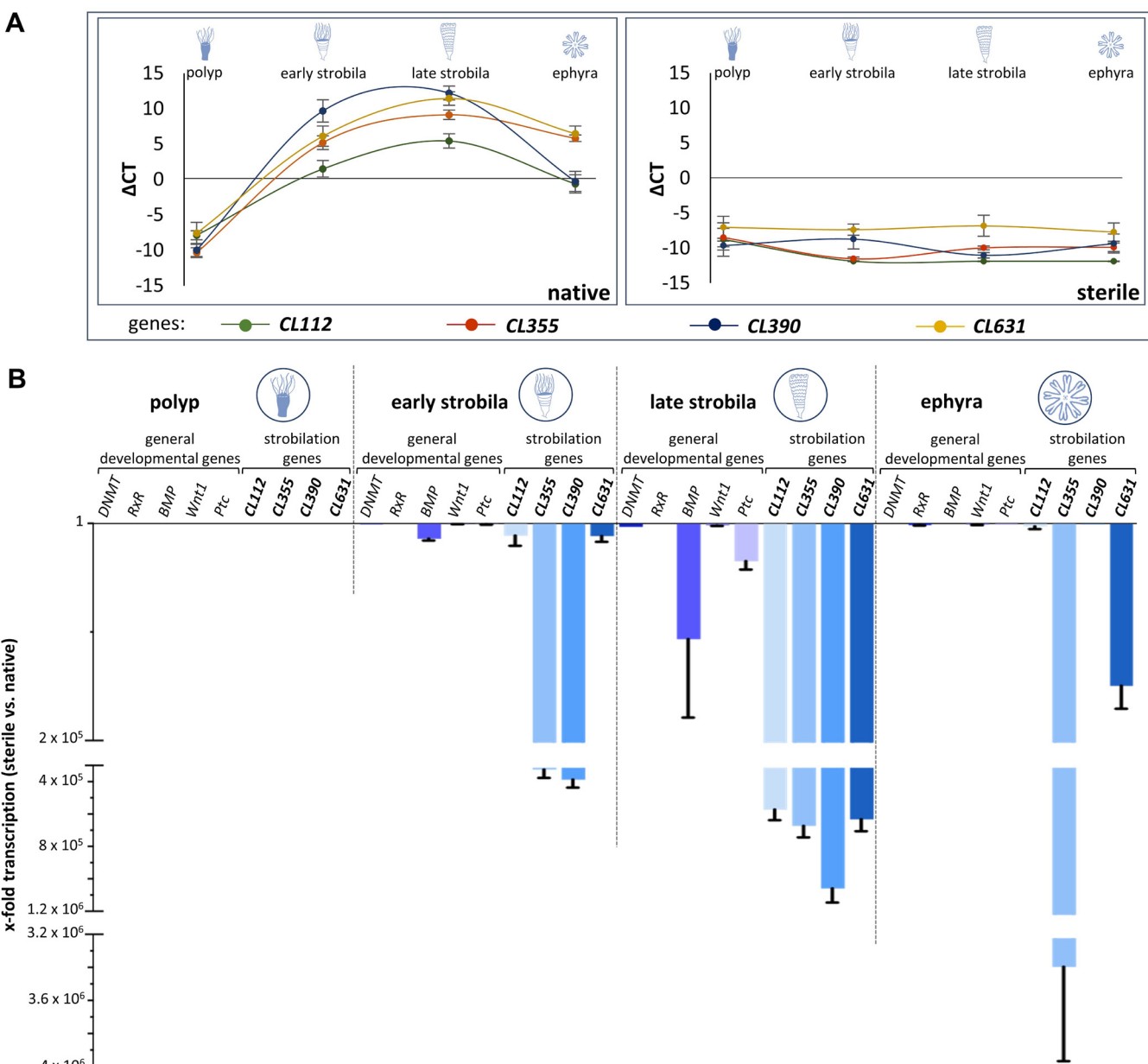

**FIG 3** Transcription of *Aurelia*-specific strobilation genes depends on presence of a microbiota. (A) Transcription patterns (reported as $\Delta C_T$ using *EF1* for normalization) of *Aurelia*-specific strobilation genes *CL112*, *CL355*, *CL390*, and *CL631* in the life stages of polyps, early strobila, late strobila, and ephyra of native (left) and sterile (right) animals. Averages of four biological replicates with three technical replicates each are reported. (B) Fold changes of transcription of selected conserved developmental genes and *A. aurita*-specific strobilation genes in sterile polyps compared to the corresponding native life stage.

**Presence of microbiota affects transcription of host developmental genes.** In order to uncover whether and how the microbial presence influences the transition of life stages within the life cycle of *A. aurita* on the molecular level, transcription levels of the four *A. aurita*-specific genes shown to be involved in strobilation (*CL112*, *CL355*, *CL390*, and *CL631*) (33, 39, 42) were compared between native and sterile animals (Fig. 3A, left panel). Since the life cycle of sterile animals is disturbed, the latter were analyzed at the time points where the normal developmental life stages would appear, although phenotypically, the sterile animals were not at that stage and showed an abnormal morphology. In the native polyp stage, none of the investigated genes was expressed, giving negative threshold cycle ($\Delta C_T$) values after normalization (Fig. 3A). Once strobilation had started, their transcription levels rose to detectable levels [$C_{T(CL112)} = 26.6$; $C_{T(CL355)} = 22.9$; $C_{T(CL631)} = 21.9$; $C_{T(CL390)} = 18.4$]. Their maximal expression was reached during the formation of late

strobilae [$C_{T(CL112)}$ = 22.6; $C_{T(CL355)}$ = 18.9; $C_{T(CL631)}$ = 16.7; $C_{T(CL390)}$ = 15.9] and decreased again during the ephyra stage (Fig. 3A, left panel). Under sterile conditions, the *A. aurita*-specific strobilation genes were not expressed at all (Fig. 3A, right panel; for $C_T$ values see Table S1 in the supplemental material).

The gene expression analysis was additionally performed with the developmental genes *DNMT*, *RxR*, *BMP*, *Wnt1*, and *Ptc*. These genes produced similar expression patterns over time during the development of native animals but were not expressed under sterile conditions (Fig. S1).

The differential gene expression levels (fold changes) between the native and respective sterile life stages are visualized in Fig. 3B for the strobilation genes as well as the conserved developmental genes. An overall downregulation was observed in sterile animals, except for the polyp stage, where none of these genes was expressed. The downregulation of the specific strobilation genes by up to a factor of $4 \times 10^6$ was notable, while the general developmental genes decreased more moderately. The transcription of all four strobilation genes was most strongly decreased in sterile late strobilae, while *CL355* and *CL390* transcription levels were also strongly decreased in sterile early strobilae ($10^5$-fold decrease). At the ephyra stage of sterile animals, the transcript level of *CL355* was still $10^6$-fold lower than that of native ephyra. Together, these data showed reduced or even absent transcription of five developmental and four strobilation-associated genes during *A. aurita* strobilation in the absence of a microbiota.

**Timing of microbial presence is crucial for polyp-to-medusa transition of *A. aurita*.** The development of the polyp, early strobila, late strobila, and ephyra stages during the recolonization treatments was monitored (Fig. 4A). The number of segments formed per animal as well as the number of ephyrae shed were compared to native conditions (Fig. 4B). Strobilation of native polyps (native conditions [NC]) is characterized by prolongation of the polyp body and absorption of tentacles, a brownish color followed by constriction, and appearance of segments. By day 5 postinduction, 100% of the native polyps (*N* = 90) had formed early strobilae (Fig. 4A, left column). Full segmentation giving late strobilae was complete by day 9, when all animals had on average 8 segments. Ephyrae were released from day 12 onwards, and all animals released on average 7 ephyrae each. As expected, the development of strobilae under sterile conditions (SC) was significantly impaired and characterized by malformations (Fig. 4A, second column). Only 52% of sterile polyps (*N* = 89) developed a strobila displaying a pale color, slim body shape, and a lack of tentacles. On average, only four segments per strobila were formed. Consequently, a significantly reduced number of released ephyrae was observed: only 17% of the animals released ephyrae at all, giving a total of 3% ephyrae at the end of the experiment (day 23 postinduction) compared to animals under NC. The individual administration of antibiotics to native polyps also reduced the native bacterial community monitored by a 30 to 80% decrease in CFU per milliliter. This reduction was concurrent with the observed phenotypic abnormalities of sterile polyps. Sterile polyps that were recolonized for 48 h following the antibiotic treatment (R1) produced phenotypes and strobilation output comparable to NC (Fig. 4A, third column), with 100% of the animals forming a strobila (*N* = 95) (with 8 segments per animal) and all individuals releasing ephyrae, with an average of 7 per animal (Fig. 4A and B). When early (R2) or late (R3) strobilae were recolonized with the native polyp microbiota, this resulted in malformed phenotypes similar to sterile polyps (Fig. 4A). Following R2 treatment, 49% of the animals (*N* = 86) showed constrictions, but the process was strongly delayed and on average of 4 segments per animal were generated. Overall, of those 49% formed strobilae, 33% released on average 1 ephyra per animal that appeared with a smaller and deformed morphology (Fig. 4A and B). Similarly, when sterile animals were recolonized at the late strobilae (R3), 48% (*N* = 84) showed strobila formation with 4 segments per animal. Finally, 10% of the 48% strobilae released one malformed ephyra per animal (Fig. 4A and B). Even the recolonization of sterile early strobila (R2esm) and late strobila (R3lsm), with their respective life stage-specific microbiota, resulted in malformed phenotypes similar to sterile polyps. Thirty-eight to 50% strobilae were formed with an average of 4 segments and no or, exceptionally, one malformed ephyra per animal was released (Fig. 4C). The observed malformation of the strobila phenotype and the strongly reduced

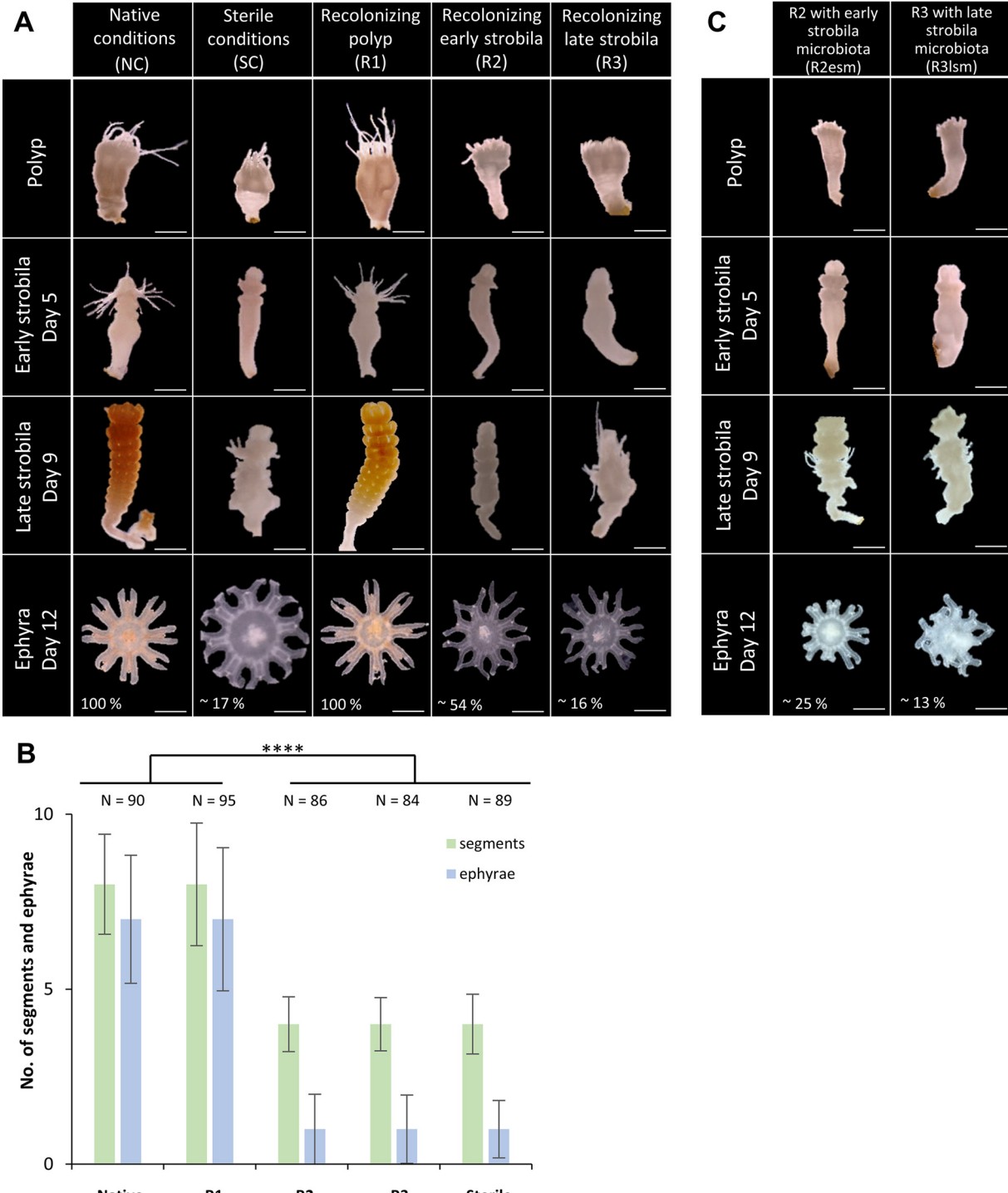

FIG 4 Timing of recolonization is crucial to restore asexual reproduction. (A) Morphology of polyps undergoing strobilation under various conditions. Left to right: native conditions (NC), sterile conditions (SC), and recolonization of sterile polyps (R1) of sterile early strobila (R2) and of sterile late strobila (R3). Photographs show the phenotypical appearance of polyps, early strobilae, late strobilae, and ephyrae (from the top). One hundred percent of the animals of NC and R1 released ephyrae, whereas 54% of R2, 16% of R3, and 17% of SC released a reduced number of ephyrae. The number of days postinfection is specified. Scale bars, 1 mm. (B) Bar plot summarizing the number of segments per animal at late strobila stage (9 days postinduction) and of released ephyrae per animal (14 days postinduction). Significance of comparisons are shown: ****, $P < 0.0001$. (C) Phenotypes of sterile polyps undergoing strobilation when sterile early strobilae (R2esm) and sterile late strobilae (R3lsm) were recolonized with their life stage-specific microbiota.

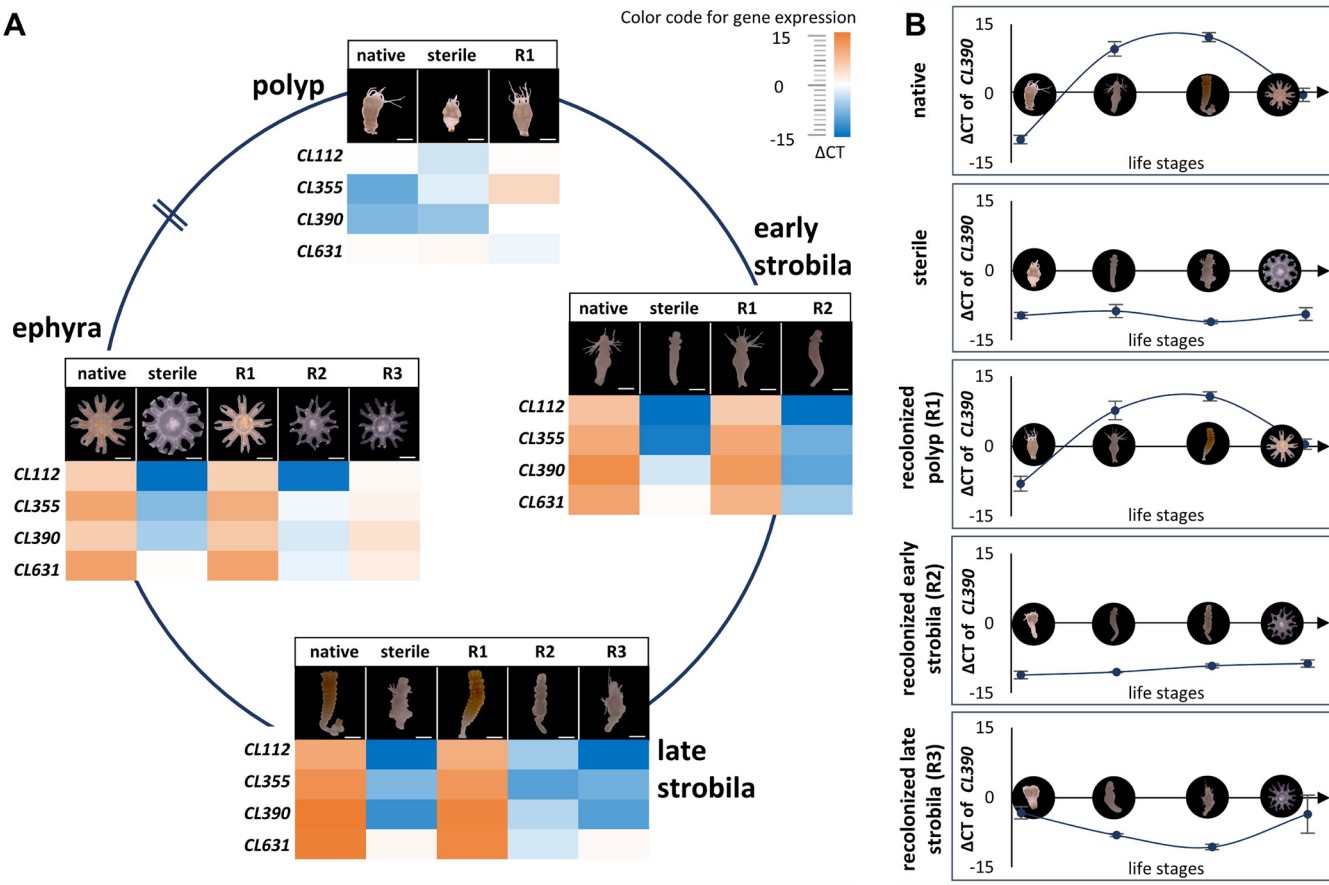

**FIG 5** Impact of the native microbiota on the transcription of *A. aurita*-specific strobilation genes during asexual reproduction. (A) Heat map of normalized $C_T$ values ($\Delta C_T$) of genes *CL112*, *CL355*, *CL390*, and *CL631* using the housekeeping gene *EF1* for normalization. $\Delta C_T$ values represent an average of four biological replicates with three technical replicates for each. (B) Expression profile of *CL390* during development over time for the different treatments. A complete summary of all assessed genes is presented in Fig. S3 in the supplemental material.

release of ephyrae following recolonization of sterile early or late strobila, but not with recolonization of sterile polyps, indicated that the timing of recolonization with a native microbiota is necessary to enable the polyp-to-medusa transition. The strobilation of sterile *A. aurita* can exclusively be enabled when recolonizing is performed at the polyp stage prior to the initiation of strobilation.

In order to exclude the possibility that the inducer, required for strobilation, had been degraded or modified by the added bacteria, treatments R2 and R3 were repeated with the constant addition of the inducer. However, these R2i and R3i treatments did not result in normal development (Fig. S2), as they also produced deformed strobilae and severely deformed ephyrae. The deformation was thus unlikely caused by insufficient inducer availability or activity, but rather by the timing of recolonization at a time point that no longer allowed restoration of natural development. The presence of microbiota was essential and required before the animals entered the strobilation process to enable the development and release of normal ephyrae.

The gene expression of the *A. aurita*-specific strobilation genes *CL112*, *CL355*, *CL390*, and *CL631* was determined by qRT-PCR for animals from all treatments at each of the four developmental stages (polyps, early and late strobilae, and ephyrae). In line with the morphological observations, native polyps and recolonized sterile polyps displayed similar expression profiles for these genes (Fig. 5A), and during development, their expression changed similarly. Transcription levels of *CL390* increased in the early strobila stage and peaked in late strobila both in native and in R1 animals (Fig. 5B). The lack of transcription of strobilation genes in sterile polyps (Fig. 3A and Fig. 5A) could not be overcome by the recolonization of sterile early and late strobilae. Recolonized early and late strobilae mirrored the transcription

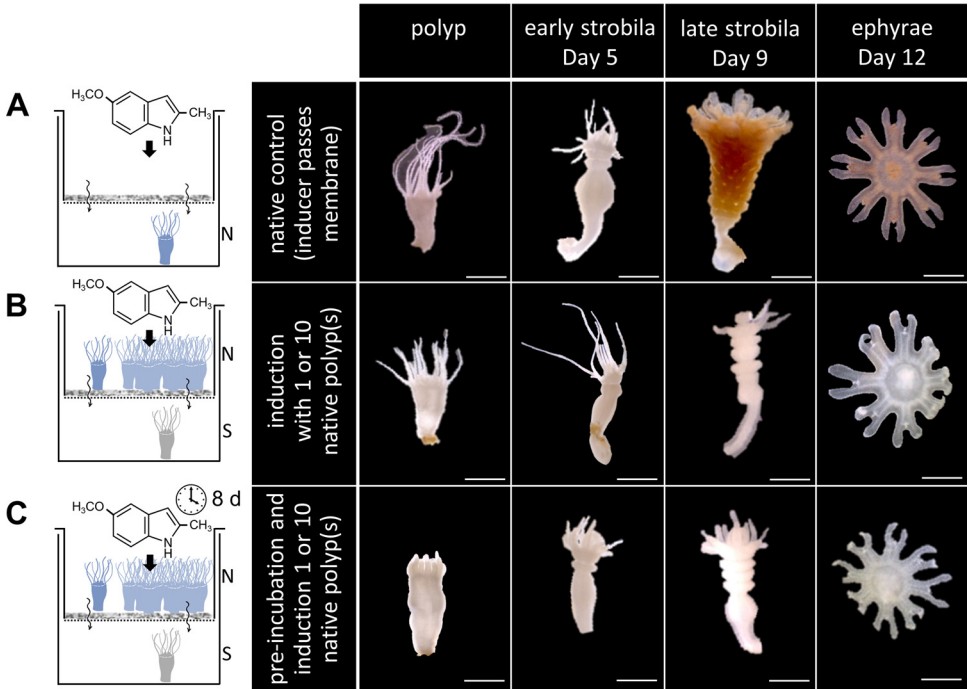

**FIG 6** Direct contact between polyp and bacteria is assumed for a normal strobilation process. Strobilation was induced with the chemical inducer that is able to pass the membrane (0.2-$\mu$m sterile filter). Scale bars, 1 mm. (A) A single native polyp was physically separated by the membrane from the strobilation inducer-ASW solution. (B) A single sterile polyp was separated by the membrane that hinders microbial cells from passing from a single polyp or a pool of 10 native polyps. Chemical inducer was present from the beginning. (C) A single sterile polyp was separated by the membrane that hinders microbial cells from passing from a single polyp or a pool of 10 native polyps. After 8 days of preincubation, strobilation was induced.

profile observed for sterile polyps (Fig. 5A). The expression patterns of *CL390* (Fig. 5B) are representative for all strobilation genes in native and recolonized polyps, as summarized in Fig. S3, where all data for all assessed genes are summarized. Consequently, the presence of a native microbiota after the onset of strobilation does not restore a normal offspring output, nor does it allow for the transcription of the strobilation genes. The developmental path of sterile animals is irreversible once the strobilation program is initiated.

To understand, if a bacterial metabolite or the metabolization of a host molecule by bacteria could be involved in initiating strobilation and whether a direct cell-to-cell contact is required, sterile polyps were incubated alongside native polyps; however, they were separated by a 0.2-$\mu$m filter that would only allow the transfer of metabolites, but not of bacteria. We observed malformed strobilae and an almost complete halt of ephyrae release of sterile polyps, whereas native polyps showed normal offspring output (Fig. 6). From this, we propose that a direct contact between host and bacteria is needed (e.g., by a type 6 secretion system) and only that leads to a healthy strobilation process. Alternatively, during this direct contact a host receptor might directly perceive a signal, due to the contact itself or via a bacterial product produced in response to the contact.

Overall, the presented findings underline that the time point of recolonization is pivotal for the further course of metamorphosis toward offspring generation of *A. aurita*. Recolonization after initiation of strobilation does not restore a healthy offspring output, most likely as a result of a now-permanently impaired expression of strobilation genes. Only recolonization of the polyp stage ensures the resetting of transcription of strobilation genes to enable a correct polyp-to-medusa transition.

## DISCUSSION

Our findings not only illustrate the importance of a native microbiome for the transition of the life stages in *A. aurita* but also demonstrate that the bacterial impact is required for normal offspring generation before the animal enters strobilation.

The absence of the microbiota and the resulting phenotypic and developmental consequences for the animal correlate with the regulation of developmental and strobilation genes (Fig. 3). Recolonization experiments on sterile animals only resulted in native expression patterns of strobilation genes, with full segmentation of the polyp and healthy offspring generation, when performed before the polyp entered the molecular strobilation process. Recolonization during the ongoing strobilation process (early and late strobila) neither molecularly nor morphologically restored this part of the life cycle. Instead, the treatments resulted in lower transcription levels, as seen in sterile polyps, as well as a malformation of the reproduction phenotype. We cannot be sure that the morphological and molecular observations of sterility were exclusively based on the absence of the native microbiome. Off-target antibiotic effects may also result in deformed *A. aurita* polyps, as has been reported for single antibiotics in various animals (46, 47). For instance, chloramphenicol negatively affected the hatching levels of eggs from the Hawaiian bobtail squid, *Euprymna scolopes* (48), though the juvenile squid and adults were not affected by similar doses of chloramphenicol (49, 50). However, mostly, authors did not conduct their experiments under the premise of the metaorganism-holobiont concept and thus regarded consequences of antibiotics without considering the microbiome as a key player, in that the effects of impairments on the animal were caused by antibiotic-induced microbiome disturbances. Since fitness traits related to strobilation can be restored after recolonization of sterile *A. aurita* polyps with the native microbiota, it is likely that the observed sterility is rather related to the microbiome than to potential toxicity of antibiotics on polyps, implying that antibiotics are the cause but not the reason for strobilation malfunctioning in polyps.

Similar observations were revealed in the cockroach *Periplaneta americana*. In the absence of its respective microbiome, the cockroach exhibits a starvation-like transcriptional response in growth and development (51). In addition, the fruit fly *Drosophila* requires its associated microbiome for reproduction, and the microbial composition and timing of this microbial presence have a significant impact on the life span of the insect (12). Recolonization of germfree *Drosophila melanogaster* embryos restored their development (52, 53), as it did with *A. aurita*. Certainly, the absence of beneficial microorganisms in the coral *Mussismilia hispida* resulted in a decreased recovery and stress attenuation processes in gene expression, promoting the effects of microbes on the host's genetic alternations (54).

In mice, the gene expression of mucus regulatory proteins differs between native and germfree animals, indicating that specific groups of bacteria accelerate the onset of development during postnatal development (55). The impact of a natural microbiota on the development of animal hosts has been observed across phyla, and the timing of its presence, which most likely includes microbiome-host interactions, is essential for natural development.

Polyps from jellyfish species, such like *A. aurita* and *Chrysaora plocamia*, harbor a specific bacterial community with a high diversity that undergoes significant restructuring during the polyp-to-medusa transition (28, 56). The specific microbial community of *A. aurita* is assumed to be essential for the sessile life stage (28). In return, the bacteria likely benefit from the polyp's mucus as a source of nutrients (28, 57, 58). We observed that polyps with naturally associated native microbiota and sterile polyps after recolonization harbored more similar microbial communities in terms of diversity than either of them with the bacterial population in the surrounding artificial seawater (Fig. 2), which corroborates with other studies (59, 60) and points to an impact of the host on the establishment and maintenance of the microbiota resulting in the same molecular and phenotypic consequences. Compared to the native counterparts, the abundance of *Vibrio* and *Colwellia* is increasing in recolonized polyps. Without the host, the inoculum in ASW shifted the bacterial population significantly to a structure of *Vibrio*, *Thalassotalea*, and *Marinicella* accompanied with a lower $\alpha$-diversity. Consequently, we assume that the host is responsible for shaping the community structure, as has been suggested by others (61). The reassembly of the microbiota on the host presumes interactions between the host and bacteria as

well as among bacteria. Host attraction and defense mechanisms (62) or bacterial interactions like quorum sensing and quorum quenching (31) could be responsible for ensuring a specific microbial community pattern of *Aurelia*. Bacteria-bacteria interactions are assumed to shape the microbiome of *Nematostella* dynamically, according to the host's developmental stage (62, 63). Such interactions can include secreted bacterial compounds that kill or impair their bacterial neighbor cells (62, 64, 65). Examples of host-bacteria interactions shaping the microbial community composition of *D. melanogaster*, *Ciona intestinales* or *Caenorhabditis elegans* have been described (66, 67). Distinct host-bacteria interactions for communication in basal metazoans are widely based on host receptors exchanging information (2). Such information affects or promotes the susceptibility to colonize a host by different molecular mechanisms, often involving pattern recognition receptors or Toll-like receptors that can recognize a wide range of microbe-derived signals (2, 68).

In our experiments, certain bacteria were missing in the microbiota of recolonized polyps compared to native polyps, indicating redundant microbiome members, or at least members not required for normal strobilation. Interestingly, recolonization increased the abundance of *Vibrio* compared to native polyps, assuming a potential function of *Vibrio* for the strobilation (Fig. 2A). *Vibrionaceae* are also found with a high abundance in corals and in the mucus of other marine invertebrates (69, 70). Nonetheless, the abundance of *Vibrio* we reported may only be a snapshot representing the situation after 2 days of recolonization and could change over time. Although, due to the long-term husbandry of the polyps, the observed abundance of the native microbiome can deviate from that of polyps in the natural ecosystem, thus partly reflecting the actual accuracy for the ecology of *A. aurita*.

The pathways of microbes modulating the host's metamorphosis or developmental processes have been reported for various animals (see above and references 71 to 73). In *A. aurita*, we demonstrated that a microbial presence is essential before the animals enter the strobilation process and that it affects the transcription of strobilation genes. This could be related to microbe-derived molecules and/or a direct host-microbe contact. Several studies have demonstrated that a bacterial metabolite or product initiates host metamorphosis (74–76). The bacterium *Pseudoalteromonas luteoviolacea* requires physical contact with the serpulid polychaete *Hydroides elegans* to produce syringe-like structures, called metamorphosis-associated contractile structures that inject stimulatory proteins crucial for the metamorphosis of the worm (77). Soluble bacteria-derived products of a yet-unknown nature have been shown to induce settlement and metamorphosis of oyster larvae (*Crassostrea gigas*) (78). To determine, whether direct contact between *Aurelia* and bacteria is required for strobilation, an experimental design with a membrane that separated sterile and native polyps was used. This ensured the absence of direct host-bacteria contact but allowed the potential transfer of molecules and enzymes to the sterile polyp (Fig. 6). Since no asexual reproduction in the latter was obtained, we predict that direct contact between host and bacteria is required and only this leads to a healthy strobilation process in *A. aurita*. It remains to be elucidated whether the polyp directly perceives the signal due to the physical contact itself, or a bacterial signal is transferred to the host in response to such contact.

In conclusion, our results revealed the regulation of strobilation-associated genes that would not occur in the absence of a polyp-associated microbiome prior to entering the strobilation process. Microbe presence and possibly interactions play a key role in the development of *A. aurita*. The present study suggests that a well-timed direct contact between the polyp and its associated bacteria enables the induction of a healthy metamorphosis and co-occurs with the presence of host genes. Whether the direct contact enables putative bacterial molecules to be transferred to the host, or alternatively, receptors that may be activated by physical contact, needs to be investigated in future experiments. Experiments with reduced manipulated communities might decipher specific bacteria and/or allow for the identification of potential interaction-related molecules. Identifying such molecules may provide insights into the microbial contribution that allow the polyp-to-medusa transition in this evolutionary ancient metazoan.

## MATERIALS AND METHODS

**Husbandry of *Aurelia aurita*.** *Aurelia aurita* polyps were derived from the North Atlantic (Roscoff, France) almost 15 years ago and have since been kept under standard husbandry conditions in ASW with a salinity of 30 practical salinity units at 19°C, as described by Weiland-Bräuer et al. (11, 79, 80). Briefly, the polyps were kept in aerated 2-liter plastic tanks under dark conditions; they were fed twice a week with freshly hatched *Artemia salina* (Hobby, Grafschaft-Gelsdorf, Germany), and the water was changed once a week with aerified ASW. Those polyps are defined as native polyps in this study.

**Isolation and characterization of *A. aurita* microbiota.** For isolation of *A. aurita*-associated stage-specific microbiota, 10 polyps before the initiation of strobilation, 10 early strobilae (5 days postinduction), and 10 late strobilae (9 days postinduction) were separately transferred into 500 $\mu$L sterile ASW and washed three times to remove transient bacteria. The animals were collectively homogenized with a motorized pestle (Kontes, DWK Life Science, Wertheim, Germany) and filtered through a 3.1-$\mu$m filter (Lab Logistic Group, Meckenheim, Germany) to remove eukaryotic cells. Aliquots ($10^5$ cells/mL) of the filtrate were immediately frozen in liquid nitrogen and stored at −80°C in 10% glycerol. These frozen stocks served as inoculum for all recolonization experiments. Cell numbers were determined by fluorescence microscopy (Axio Scope microscope and Axio Vision software; Zeiss, Jena, Germany) using the nucleic acid staining dye SYTO9 of the LIVE/DEAD BacLight bacterial viability kit (1:1,000; Invitrogen, Darmstadt, Germany) and a Neubauer count chamber (Assistant, Sondheim vor der Röhn, Germany) according to the manufacturer's instructions. The absolute number of cells per polyp was calculated as $2.8 \times 10^6$ cells for one single polyp.

Total bacterial DNA was extracted from thawed inoculum using the Wizard Genomic DNA purification kit (Promega), and the V1 and V2 region of the 16S rRNA gene were amplified as described by Weiland-Bräuer et al. (11) using 27 F/338 R primers (27 F, 5′-AGAGTTTGATCCTGGCTCAG-3′; 338 R, 5′-TGCTGCCTCCCGTAGGAGT-3′). All primers contained unique identifier sequences (barcodes) to distinguish between the samples following published methods (81) with an optimized primer design (82). Amplification was performed in 20 $\mu$L using Phusion high-fidelity DNA polymerase (New England Biolabs) with a protocol of 98°C, 30 s followed by 30 cycles (98°C, 9 s; 55°C, 60 s; 72°C, 90 s), 10-min extension at 72°C, and 10-min 10°C cooling. The same procedure was used to characterize the microbiota of six replicates of homogenized polyps or recolonized animals (see below).

All amplicons were purified using the MiniElute gel extraction kit (Qiagen, Hilden, Germany), quantified using the Quant-iT PicoGreen kit (Invitrogen, Darmstadt, Germany), and sequenced on an Illumina platform, for which equal amounts were pooled.

Paired-end raw read files were processed using QIIME 2 (version qiime2-2021.2 [83]). The 16S rRNA gene sequences were denoised using DADA2 via q2-dada2. Quality profile plots were inspected to select values for truncation of forward and reverse reads. The truncation and trimming were set to "–p-trim-left-f 13, –p-trim-left-r 13; and –p-trunc-len -f 250, –p-trunc-len-r 250." All resulting amplicon sequence variants (ASVs) were aligned with mafft via q2-, and this was used to construct a midpoint-rooted phylogeny with fasttree2. Bacterial taxonomic assignment was done using feature-classifier classify-consensus-vsearch against the Silva 132 v. 99 QIIME2-compatible database. $\alpha$-Diversity metrics based on observed ASVs were estimated using q2-diversity after samples were rarefied (subsampled without replacement) to 2.106 sequences per sample to determine differences among the bacterial communities driven by different treatments.

**Generation of germfree polyps.** Preliminary experiments with single and mixed antibiotics (chloramphenicol, neomycin, ampicillin, streptomycin, rifampin [each at 50 mg/liter] and 60 mg/liter spectinomycin) were performed to establish the final protocol for the generation of sterile polyps. The remaining cultivable bacteria were analyzed based on CFU per milliliter on MB and R2A agar plates after 5 days of incubation. Consequently, sterile polyps were generated by treatment of 3-day-starved animals with an antibiotic mixture containing 50 mg/liter each of chloramphenicol, neomycin, ampicillin, streptomycin, and rifampin, and 60 mg/liter spectinomycin (all from Carl Roth, Karlsruhe, Germany) in sterile ASW. After 5 days, absence of cultivable bacteria was ensured by plating polyp homogenate on MB and R2A agar plates. In addition, the genomic DNA of the polyps was used as the template for full-length 16S rRNA gene amplification using GoTaq polymerase with universal primers 27F (5′-AGAGTTTGATCCTGGCTCAG-3′) and 1492R (5′-GGTTACCTTGTTACGACTT-3′). The absence of amplicons in combination with the absence of colonies confirmed that the polyp, early strobila, and late strobila stages were sterile.

**Recolonization of sterile *A. aurita* at different developmental stages.** Single sterile polyps in individual wells of a 48-well plate containing 1 mL sterile ASW were recolonized at three developmental stages with 96 replicates per treatment. Strobilation was induced with 5 $\mu$M 5-methoxy-2-methyl indole (Carl Roth, Karlsruhe, Germany) for 3 days. From day 4, the inducer was omitted (unless stated otherwise) and developing segments and the release of ephyrae were recorded daily using a stereomicroscope (Novex Binoculares RZB-PL Zoom-Microscope 65.500, Novex, Arnhem, the Netherlands) equipped with a high-definition multimedia interface camera. The stage of early strobilation was determined by the appearance of the first strobila, as detected by the constriction of the first segments at the apical part, and early strobilation was typically complete on day 5 under NC. The stage of late strobilation was reached when segmentation was complete (at day 9 for NC), while the sequential release of ephyrae was observed from day 12 onwards. The release of ephyrae was monitored until the end of the experiment on day 23.

Recolonization of sterile polyp and early and late strobila was conducted by addition of 10 $\mu$L of the thawed inoculum to ASW ($10^5$ cells/mL and per animal [see above]). Fresh ASW containing inoculum was applied the next day, so that the animals were exposed to the inoculum for 48 h. A control inoculum was incubated for 48 h in ASW only. Treatment group R1 (sterile polyps) were recolonized prior to the chemical induction. Treatment group R2 was recolonized at the early strobila stage, and treatment

**TABLE 1** Primers used for qRT PCR analysis of 10 target genes

| Gene | Primer name | Sequence (5′–3′) | Length (bp) | $T_m$ (°C) | Gene function | Reference |
|---|---|---|---|---|---|---|
| EF1 | EF1_F2 | AGTTCCAGGGGACAATGTTGG | 21 | 59.8 | Protein biosynthesis | 33 |
|  | EF1_R2 | TGGATTTCGCCAGGATGGTTC | 21 | 59.8 |  |  |
| CL112 | aCL112_F2 | GAAGCTACCAGATCCGTTTG G | 21 | 59.8 | Induction of strobilation | 33 |
|  | aCL112_R2 | TGCAAGCGCATCTGTTCACAG | 21 | 59.8 |  |  |
| CL355 | CL355_for | TTCCGGAGAGCAGACCAATG | 20 | 59.4 | Induction of strobilation | 42 |
|  | CL355_rev | CCAACGGCTGCATATACCATC | 21 | 59.8 |  |  |
| CL390 | CL390_F2 | AAGGTGCGACAATGAAGGTCC | 21 | 59.8 | Induction of strobilation | 33 |
|  | CL390_R2 | GTCTACAGGCTCAATGGTGTC | 21 | 59.8 |  |  |
| CL631 | aCL631_F2 | GCCTTGACGGTGAAAGATGAG | 21 | 59.8 | Induction of strobilation | 33 |
|  | aCL631_R2 | ACCTCGTCCTCATCCTTTTCG | 21 | 59.8 |  |  |
| DNMT | aDNMT_F2 | CATGCACAGTTCATCGTCGAG | 21 | 59.8 | Role in methylation of DNA | 33 |
|  | aDNMT_R2 | CAATCGAACGCCACGTAATGC | 21 | 59.8 |  |  |
| RxR | aRXR_Q_F | AGATGTACAGTGCCACGTTGG | 21 | 59.8 | Induction of strobilation | 33 |
|  | aRXR_Q_R | CAAGTGTTCGAGTGCTTGCAG | 21 | 59.8 |  |  |
| BMP | qBMP_F | GCAATGTAGCCCACTGCTTG | 20 | 59.4 | Secreted morphogen for cell differentiation and division | 39 |
|  | qBMP_R | ATGCTTGGCCTTGGTCGAT | 19 | 56.7 |  |  |
| Wnt1 | Wnt1_F | AGATTTCTGGCGCAAAAGGC | 20 | 57.3 | Morphogen for localization of anterior-posterior axis of strobilation segments | 42 |
|  | Wnt1_R | TGTTTGCCTGCACCATTCAG | 20 | 57.3 |  |  |
| Ptc | Ptc_for | AGTGACAGCTCAACGAGTGG | 20 | 59.4 | Receptor in Shh signal pathway | 42 |
|  | Ptc_rev | ACTCCGAGCAATGCTAACCA | 20 | 57.3 |  |  |

group R3 was recolonized at the late strobila stage. Native animals (NC) and a sterile control (SC) were included. For treatments R2 and R3, we assessed polyps with the constant presence of the inducer, giving R2i and R3i, respectively, to exclude the lasting effect of the inducer given in R1. To consider the possibility that a life stage-specific microbiota is required for normal progeny, output recolonization of sterile early strobila (R2esm) and late strobilae (R3lsm) was conducted by adding 10 μL of the thawed inoculum of the respective life stages to ASW ($10^5$ cells/mL and per animal). Effects of the cryoprotectant glycerol were excluded before the experimental start. Using the same concentrations as in the recolonization experiment (10 μL of 10% glycerol in 1 mL ASW), we incubated native and sterile polyps for 1 to 7 days in the presence of glycerol. The polyp phenotypes and survival rates were not affected.

Preliminary experiments with eight replicates for identifying a metabolite of bacterial origin or the metabolization of a host molecule by bacteria without direct contact were conducted with polyps in 1 mL sterile ASW in single 48-well microtiter plate cavities. A single sterile polyp was settled at the bottom of each cavity. A 0.2-μm centrifugal sterile filter unit (amchro, Hattersheim, Germany) was inserted into the cavity, and 10 native polyps were added to the unit. Strobilation was induced after 8 days post-addition of native polyps using 5 μM 5-methoxy-2-methyl indole (this inducer can pass the filter). Segmentation and ephyrae release were recorded for 20 days.

**Expression analysis of developmental genes by qRT-PCR.** Expression analysis of host genes was performed with 6 replicates of four development stages (polyp, early strobila, late strobila, and ephyra) for the 5 treatments (NC, SC, R1, R2, and R3). A total of 9 genes were assessed (Table 1) (33, 42). For RNA extraction, animals were transferred to 1.5-mL reaction tubes, washed three times with sterile ASW, frozen in liquid nitrogen, and stored at −80°C until use. Thawed animals were homogenized, and total RNA was isolated by phenol-chloroform extraction as follows: 200 μL of 100 mM Tris-HCl (pH 5.5), 10 mM EDTA, 0.1 M NaCl, 1% SDS, 1% 2-mercaptoethanol, and then 2 μL proteinase K (25 mg/mL; Thermo Fisher Scientific, Darmstadt, Germany) was added to the homogenate and incubated for 10 min at 55°C. Following protein precipitation with 5 μL 3 M potassium acetate, 250 μL ROTI Aqua-P/C/I (Carl Roth, Karlsruhe, Germany) for 15 min on ice, and centrifugation (14,000 rpm for 15 min, 4°C) the RNA was precipitated by adding 1 volume of 2-propanol. RNA pellets were washed once with 70% ethanol and finally dissolved in 30 μL RNase-free water (Carl Roth, Karlsruhe, Germany).

Prior to reverse transcription, DNA was removed by DNase I treatment (Thermo Fisher Scientific, Darmstadt, Germany). Successful DNA removal was ensured by standard PCR analysis using the GoTaq PCR kit (Promega, Madison, WI, USA) with primers 27F and 1492R. RNA was reverse transcribed using random hexamers of the SuperScript IV first-strand synthesis system (Invitrogen). cDNA was purified using the PCR Clean-Up kit (Macherey-Nagel, Düren, Germany).

The transcripts of 10 selected target genes were analyzed by qPCR in 25 μL Platinum SYBR green qPCR SuperMix-UDG (Invitrogen) with 5 μL cDNA (20 ng) and 0.4 μM forward and reverse primers, as listed in Table 1. The cycling conditions were 50°C for 2 min followed by 95°C for 2 min and 40 cycles of 95°C for 15 s; 60°C for 30 s; 95°C for 15 s, and then 60°C for 30 s and 95°C for 15 s. Average cycle threshold ($C_T$) values of all samples were normalized using transcripts of elongation factor 1 (EF1 [33]) as the reference [$C_{T(EF1)}$ = 20.09 ± 1.35]. Fold changes of gene regulation were calculated using the $\Delta\Delta C_T$ method (84).

Student's *t* test was used to determine statistical significance between the mean $C_T$ values per gene per group. *P* values were adjusted using the Bonferroni method to account for multiple comparisons. These calculations were performed in R.

**Data availability.** All data generated during this study are included in the published article (and its supplemental file).

Sequence data were deposited under NCBI BioProject PRJNA896887, comprising locus tag prefixes SAMN31571268 to SAMN31571288.

## SUPPLEMENTAL MATERIAL

Supplemental material is available online only.

**SUPPLEMENTAL FILE 1**, PDF file, 0.7 MB.

## ACKNOWLEDGMENTS

This work was conducted with the financial support of the DFG as part of the CRC1182 "Origin and function of metaorganisms" (Project B2).

For next-generation deep sequencing, we thank Sven Künzel and colleagues from the Department for Evolutionary Genetics of the Max Planck Institute for Evolutionary Biology.

We are further grateful for the fruiting discussions and support during the writing process by Trudy M. Wassenaar.

N.W.-B. and R.A.S. designed the research; N.J. performed the research with support from S.J.; C.M.C. analyzed the sequence data; N.J., N.W.-B., and R.A.S. wrote the paper; N.W.-B. and R.A.S. edited the paper.

We declare no competing interests.

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
