## [Reviewer comments · Microbiology Spectrum]

Microbiology Spectrum

The life cycle of *Aurelia aurita* depends on the presence of a microbiome in polyps prior to the onset of strobilation

Nadin Jensen, Nancy Weiland-Bräuer, Shindhuja Joel, Cynthia Chibani, and Ruth Schmitz

Corresponding Author(s): Ruth Schmitz, Christian-Albrechts-Universität zu Kiel

Review Timeline:

Submission Date:	January 16, 2023
Editorial Decision:	March 21, 2023
Revision Received:	April 12, 2023
Accepted:	June 6, 2023

Editor: Kevin Theis

Reviewer(s): Disclosure of reviewer identity is with reference to reviewer comments included in decision letter(s). The following individuals involved in review of your submission have agreed to reveal their identity: Janja Ceh (Reviewer #3)

Transaction Report:

DOI: <https://doi.org/10.1128/spectrum.00262-23>

March 21, 2023

Prof. Ruth Anne Schmitz-Streit
Christian-Albrechts-Universität zu Kiel
Institute for General Microbiology
Am Botanischen Garten 1-9
Kiel 24118
Germany

Re: Spectrum00262-23 (Asexual reproduction of *Aurelia aurita* depends on the presence of a balanced microbiome at polyp stage)

Dear Prof. Ruth Anne Schmitz-Streit:

Thank you for submitting your manuscript to Microbiology Spectrum. It has now been reviewed by three experts in the discipline. Their comments are included at the end of this email. If you do not receive reviewer #3's attachment with comments, please let me know. In general, the reviews are positive, but modifications are required for publication. Please let me know if you have any questions.

Link Not Available

Sincerely,

Kevin R. Theis

Journals Department
Reviewer comments:

Reviewer #1 (Public repository details (Required)):

The authors provide an NCBI project ID, but I didn't find it when I searched, so it may need to be made public prior to publication.

Reviewer #1 (Comments for the Author):

In this article Jensen and colleagues explore the impact of the complex microbiome on strobilation of the jellyfish *Aurelia aurita*. They find that the presence of the native microbiome is required at the polyp stage for normal strobilation and ephyra release, and that introduction of the microbiome at subsequent timepoints is insufficient for metamorphosis. This paper is well thought-out with very nice figures. Understanding of host-microbe interactions in complex mutualisms is an important topic and this paper advances that understanding. I have some comments (see attached) that I think should be addressed, but consider them fairly minor.

In this article, Jensen and colleagues explore the impact of the complex microbiome on strobilation of the jellyfish *Aurelia aurita*. They find that the presence of the native microbiome is required at the polyp stage for normal strobilation and ephyra release, and that introduction of the microbiome at subsequent timepoints is insufficient for metamorphosis. This paper is well thought-out with very nice figures. Understanding of host-microbe interactions in complex mutualisms is an important topic and this paper advances that understanding. I have some comments (see below) that I think should be addressed, but consider them fairly minor.

General Comments:

- 1) I'd briefly explain around line 142 that the inoculum is from filtered homogenate.
- 2) I liked your experiment demonstrating that cell-cell contact is required. My question building on this is whether you think tissue colonization by the bacteria is required, or whether you think bacteria in the water would be sufficient?
- 3) For lines 302-303 I would note that Kerwin et al. 2019 in mBio did find that chloramphenicol negatively impacted development of the Hawaiian bobtail squid embryo, but that other antibiotics did not, and that paper was looking at microbiome functionality. Not asking you to cite it, but wanted to point it out. :) I was a little confused by your description of the antibiotics tested in lines 451-458 - which were tested singly, and did those not work sufficiently? Given the known toxicity of chloramphenicol that you cite, it may have been better not to include that. I think though that since your recolonized sterile polyps were fine, the antibiotic treatment wasn't a problem.
- 4) The NCBI BioProject ID did not appear when searched - assume this will be public by publication.
- 5) Did you do a glycerol control since your inoculum were frozen with glycerol? I'm assuming not, in which case you should note that somewhere and the reasoning behind that decision.

Figure Comments:

- 1) In Figure 1 I found it unclear what was happening in the sterile pathway of A - what do the 2nd and 3rd grey images represent? A description in the legend or in your text (around line 133) would be helpful (I see you get to this later on pg 9, but I think a brief description early on would still be helpful). The NC and SC abbreviations should be defined in the legend. Does the R1 ephyra not have a question mark because that was studied in your last paper? I was unclear why the inducer was shown outside of the box for R1 and why that was different with it being inside the box in R2/R3. Also why is it R1 and not R1i if the inducer was used? Should it say inducer instead of inductor in the figure?
- 2) I really liked your color scheme for Figure 2 and the way you organized the taxa. For the bolded genera it would be good to define what your threshold was for "relative high abundance". The scroll over with mouse click functionality wasn't working for me but I assume that will be checked in the proof stage. I was surprised to see that the alpha diversity of the recolonized polyps was so much higher. This should be discussed more - do you have a hypothesis? It doesn't seem like you should be able to strongly increase in richness between what's being inoculated and what's being colonized - unless you suspect an increase in richness (which at least to me doesn't seem super apparent). You later call the diversity comparable (lines 327-329) which seems at odds with your data. Visually comparing the inoculum and the recolonized polyps I wondered if any taxa were missing - might be helpful to underline or otherwise indicate taxa present in the inoculum but missing in recolonized, or alternatively present in native but missing in recolonized.
- 3) Figure 3 is really nicely done - great job!
- 4) For part A of Figure 4 are you inducing to get ephyra in the SC, R2, R3 conditions? Or do you still get some ephyra, just a smaller percentage? The percentages in the ephyra boxes should be mentioned in the legend. For the late strobila, where is the color coming from in NC and R1? It makes it a little hard to compare to the SC and R2 since those appear more as silhouettes, but maybe that's just what they look like? (Sorry I'm not familiar with what they should look like in this species.)
- 5) Figure 5 is really nice - great presentation of complex data! How do you explain the

difference in expression between the native and R1 at the polyp stage? The figure seems to contrast with your results at lines 253-254?

6) Figure 6 refers to A/B/C but these labels aren't present in the figure itself. For B in the legend you don't mention the inducer but the figure shows it - I'd reword to make the legend clear. You need to define the scale bars in the legend.

Minor Comments:

- 1) In lines 26, 35, 46, 77, 238, 324 would it be better to say polyp-to-medusa?
- 2) In line 28 I think it should be accompanied by instead of accompanied with.
- 3) In line 65 I'm not sure whether the sentence starting Several hosts is necessary. I'm unclear as to what you mean here (unless it really is just that many different systems can be models?) so I'd either rephrase or drop it all together.
- 4) In line 95 should be The expression, not These expression.
- 5) In line 97 should be "and is expressed" not "and be expressed".
- 6) In line 265 should it be "path" instead of "faith"? Or maybe "fate"?
- 7) Sentence starting The abundance in lines 331-334 needs to be revised for clarity.
- 8) Sentence starting Examples of in lines 341-342 is quite vague and could be revised to briefly provide the actual examples.
- 9) I don't think oyster should be italicized in line 369 - might actually be better to provide the specific species.
- 10) Line 369 has a typo - should be whether, not wether.
- 11) Line 481 - revise sentence

Reviewer #2 (Comments for the Author):

Review for manuscript entitled Asexual reproduction of *Aurelia aurita* depends on the presence of a balanced microbiome at polyp stage by Nadin Jensen, Nancy Weiland-Bräuer, Shindhuja Joel, Cynthia Maria Chibani, Ruth Anne Schmitz considered for publication at Microbiology Spectrum journal.

The research of host-microbiota interactions is very up-to-date and in particular this study as it brings new insights in novel model metaorganism, i.e., jellyfish, which is unlike other systems, e.g., corals, hydrozoans, etc, less studied metazoan host-microbiota system. Being ecologist my major concern, as always with model organisms that have been capture/kept in the lab for a long period of time, is the actual accuracy or relevance of results for ecology of this organisms. In my modest opinion these systems, like the case here, when organisms have been kept in the lab for over 10 years, are far away from their natural representatives. Else, study is very well designed, conducted and analysis is very appropriate and paper well structured (however a bit long and repetitive in some places and hence could be more condensed and streamlined). Please see my specific comments below.

Some minor comments:

Abstract:

Line 23: Yet, the timing and molecular consequences of the microbiome during the 24 strobilation process had not been investigated. Is strangely formulated - the timing and molecular consequences of the microbiome- please re-write.

Introduction:

Line 86-87: Several proteins, including transcriptional regulators have been shown to be involved in the induction of strobilation. Reference missing.

Line 97-98: These expression of these four genes is strongly upregulated in polyps entering the strobilation phase. CL112 97 encodes a secreted protein containing an epidermal growth factor-like domain and be expressed in endodermal cells of each segment of the early strobili. Reference missing

Results/Methods:

General comment: can you please justify use of V1-V2 region? Some microbiota could be missed in this case. Did you try other primer sets and compare?

Methods:

Line 406:

For isolation of *A. aurita*-associated microbiota, 10 polyps, 10 early strobilae (5 days 407 post-induction), and 10 late strobilae (9 days post-induction) were transferred into 408 500 µl sterile ASW and washed three times to remove transient bacteria. The animals were collectively homogenized with a motorized pestle (KONTES, DWK Life Science, Wertheim, Germany) and filtered through a 3.1 µm filter (Lab Logistic group, Meckenheim, Germany) to remove eukaryotic cells. This is not clear to me. Did you combine/pool microbiota from all this different 'hosts' to obtain inoculum or you kept it separately for each 'host'? Depending on

what exactly you did in my opinion affects the outcome substantially.

Line 412: Aliquots (105 cells/ml) of the filtrate were immediately frozen in liquid nitrogen and stored at -80 {degree sign}C in the presence of 10 % glycerol. These frozen stocks served as inoculum for all recolonization experiments presented here. I wonder how many of cells lost their viability via this process - maybe also specific taxa...did you maybe check the effect on microbial community?

Line 482: Recolonization of sterile polyps, early and late strobila was conducted by addition of 10 µl of the thawed inoculum to ASW (105 cells/ml and per animal, see above), with fresh ASW containing inoculum applied the next day, so that the animals were exposed to the inoculum for 48 h. A control of inoculum was incubated for 48 h in ASW only. At this point I started to think about viable/dead cells within inoculum and then I realized that you used STY00 to count - which allows you to distinguish live/dead cells, however maybe not all readers know that so would be nice to specify this somewhere..else makes you wonder how much viability of cells was affected by your freezing/thawing of inoculum, hence also affecting your community composition, Results:

Line 149: what about alpha diversity of whole community?

Line 152 (and other cases): when you talk about relative abundance one needs to be aware that of course this depends on total cell abundance, e.g., one taxa can increase in relative abundance, but overall cell abundance (cells/mL) is lower and this is important!, also an increase/decrease can be due to decrease/increase of other taxa...it can give a wrong impression! Since I understood you used SYTO9 to distinguish between live and dead and made an effort to count cells on polyps etc., why not making an effort to take this actual cell numbers into account

Line 156-157: sentence not clear, please re-write: Overall, the microbiota of recolonized polyps more strongly resembled that of native polyps than bacteria maintained in artificial seawater (ASW) in absence of animals.

Line 216-216: The individual administration of antibiotics also reduced the native bacterial community monitored by a decrease of 30 - 80 % in CFU/ml. This is confusing to me: when you talk about CFU/mL I assume only cultivable part of community was assessed? But to my understanding this was not the case. Please explain.

Line 270-272: For this, native polyps with their diverse, healthy microbiome were incubated together with sterile polyps, but separated by a 0.2 µm filter that allowed the transfer of metabolites, but not of bacteria. If we are picky we would take into account the fact that some really small bacteria can escape 0.2 um filtration!

Discussion:

Line 287: Our results not only illustrate how important a native microbiome is for the asexual reproduction of *A. aurita*, but also demonstrated that the generation of offspring is only possible when the microbiome is present before the animal enters the strobilation process. Just a general thought: I cannot image where and when in natural scenario microbiome would not be present before animal enters strobilation process, so not really sure, from ecological perspective, why this I even relevant. Maybe discussion on this aspect could be elaborate.

Line 332-334: The abundance of *Vibrio* and *Colwellia* is increasing in recolonized polyps, whereas ASW shifted the bacterial population to a significantly structure accompanied with a lower α -diversity. This is a very strange sentence, I don't understand what is meant here. Please correct.

Line 336-337: During a reassembly of the microbiota on the host's mucus, interactions between host and bacteria as well as amongst bacteria ensure a specific microbial community pattern of *Aurelia*. How can you possibly know this? Did you test other jellyfish species and compare species-specificity of associated microbiota?

Reviewer #3 (Comments for the Author):

Dear authors, this is an interesting study, however, the manuscript needs a lot of work; the good news - there is nothing that can't be fixed (see comments in the manuscript file). The original title is misleading, general concepts of the jellyfish lifecycle are either not well understood or not precisely worded, the abstract is confusing and lacks any description of methods, there is a lack of understanding which parts belong into which section of the paper, referencing is done in a sloppy way - it's not enough to reference a study, it also needs to be put into context. There are conclusions that are not supported by the results. There is an overuse of specific words, e.g. the word crucial appears about 15 times. While the results are worth publishing, the importance of the study is generally overstated and the wording is disproportionally strong and often exaggerated to a point where it does not reflect the findings. The shortcomings of the study need to be elaborated on (e.g. husbandry conditions). Many sentences are highly confusing, thoughts need to be ordered and sentences reworded. There are grammatical and style issues and I do recommend a proper revision of the English, if not by a native speaker at least by an online grammar and punctuation checker. I feel that the manuscript has been submitted either prematurely, or in a rush, which is kind of disrespectful to the reviewers, please consider this in your future submissions. For more details see the manuscript file.

Staff Comments:

Preparing Revision Guidelines

To submit your modified manuscript, log onto the eJP submission site at <https://spectrum.msubmit.net/cgi-bin/main.plex>. Go to Author Tasks and click the appropriate manuscript title to begin the revision process. The information that you entered when you

first submitted the paper will be displayed. Please update the information as necessary. Here are a few examples of required updates that authors must address:

Please return the manuscript within 60 days; if you cannot complete the modification within this time period, please contact me. If you do not wish to modify the manuscript and prefer to submit it to another journal, please notify me of your decision immediately so that the manuscript may be formally withdrawn from consideration by Microbiology Spectrum.

In this article, Jensen and colleagues explore the impact of the complex microbiome on strobilation of the jellyfish *Aurelia aurita*. They find that the presence of the native microbiome is required at the polyp stage for normal strobilation and ephyra release, and that introduction of the microbiome at subsequent timepoints is insufficient for metamorphosis. This paper is well thought-out with very nice figures. Understanding of host-microbe interactions in complex mutualisms is an important topic and this paper advances that understanding. I have some comments (see below) that I think should be addressed, but consider them fairly minor.

General Comments:

- 1) I'd briefly explain around line 142 that the inoculum is from filtered homogenate.
- 2) I liked your experiment demonstrating that cell-cell contact is required. My question building on this is whether you think tissue colonization by the bacteria is required, or whether you think bacteria in the water would be sufficient?
- 3) For lines 302-303 I would note that Kerwin et al. 2019 in mBio did find that chloramphenicol negatively impacted development of the Hawaiian bobtail squid embryo, but that other antibiotics did not, and that paper was looking at microbiome functionality. Not asking you to cite it, but wanted to point it out. :) I was a little confused by your description of the antibiotics tested in lines 451-458 – which were tested singly, and did those not work sufficiently? Given the known toxicity of chloramphenicol that you cite, it may have been better not to include that. I think though that since your recolonized sterile polyps were fine, the antibiotic treatment wasn't a problem.
- 4) The NCBI BioProject ID did not appear when searched – assume this will be public by publication.
- 5) Did you do a glycerol control since your inoculum were frozen with glycerol? I'm assuming not, in which case you should note that somewhere and the reasoning behind that decision.

Figure Comments:

- 1) In Figure 1 I found it unclear what was happening in the sterile pathway of A – what do the 2nd and 3rd grey images represent? A description in the legend or in your text (around line 133) would be helpful (I see you get to this later on pg 9, but I think a brief description early on would still be helpful). The NC and SC abbreviations should be defined in the legend. Does the R1 ephyra not have a question mark because that was studied in your last paper? I was unclear why the inducer was shown outside of the box for R1 and why that was different with it being inside the box in R2/R3. Also why is it R1 and not R1i if the inducer was used? Should it say inducer instead of inductor in the figure?
- 2) I really liked your color scheme for Figure 2 and the way you organized the taxa. For the bolded genera it would be good to define what your threshold was for “relative high abundance”. The scroll over with mouse click functionality wasn't working for me but I assume that will be checked in the proof stage. I was surprised to see that the alpha diversity of the recolonized polyps was so much higher. This should be discussed more – do you have a hypothesis? It doesn't seem like you should be able to strongly increase in

richness between what's being inoculated and what's being colonized – unless you suspect an increase in richness (which at least to me doesn't seem super apparent). You later call the diversity comparable (lines 327-329) which seems at odds with your data. Visually comparing the inoculum and the recolonized polyps I wondered if any taxa were missing – might be helpful to underline or otherwise indicate taxa present in the inoculum but missing in recolonized, or alternatively present in native but missing in recolonized.

- 3) Figure 3 is really nicely done – great job!
- 4) For part A of Figure 4 are you inducing to get ephyra in the SC, R2, R3 conditions? Or do you still get some ephyra, just a smaller percentage? The percentages in the ephyra boxes should be mentioned in the legend. For the late strobila, where is the color coming from in NC and R1? It makes it a little hard to compare to the SC and R2 since those appear more as silhouettes, but maybe that's just what they look like? (Sorry I'm not familiar with what they should look like in this species.)
- 5) Figure 5 is really nice – great presentation of complex data! How do you explain the difference in expression between the native and R1 at the polyp stage? The figure seems to contrast with your results at lines 253-254?
- 6) Figure 6 refers to A/B/C but these labels aren't present in the figure itself. For B in the legend you don't mention the inducer but the figure shows it – I'd reword to make the legend clear. You need to define the scale bars in the legend.

Minor Comments:

- 1) In lines 26, 35, 46, 77, 238, 324 would it be better to say polyp-to-medusa?
- 2) In line 28 I think it should be accompanied by instead of accompanied with.
- 3) In line 65 I'm not sure whether the sentence starting Several hosts is necessary. I'm unclear as to what you mean here (unless it really is just that many different systems can be models?) so I'd either rephrase or drop it all together.
- 4) In line 95 should be The expression, not These expression.
- 5) In line 97 should be “and is expressed” not “and be expressed”.
- 6) In line 265 should it be “path” instead of “faith”? Or maybe “fate”?
- 7) Sentence starting The abundance in lines 331-334 needs to be revised for clarity.
- 8) Sentence starting Examples of in lines 341-342 is quite vague and could be revised to briefly provide the actual examples.
- 9) I don't think oyster should be italicized in line 369 – might actually be better to provide the specific species.
- 10) Line 369 has a typo – should be whether, not wether.
- 11) Line 481 – revise sentence

Review for manuscript entitled Asexual reproduction of *Aurelia aurita* depends on the presence of a balanced microbiome at polyp stage by Nadin Jensen, Nancy Weiland-Bräuer, Shindhuja Joel, Cynthia Maria Chibani, Ruth Anne Schmitz considered for publication at *Microbiology Spectrum* journal.

The research of host-microbiota interactions is very up-to-date and in particular this study as it brings new insights in novel model metaorganism, i.e., jellyfish, which is unlike other systems, e.g., corals, hydrozoans, etc, less studied metazoan host-microbiota system. Being ecologist my major concern, as always with model organisms that have been capture/kept in the lab for a long period of time, is the actual accuracy or relevance of results for ecology of these organisms. In my modest opinion these systems, like the case here, when organisms have been kept in the lab for over 10 years, are far away from their natural representatives. Else, study is very well designed, conducted and analysis is very appropriate and paper well structured (however a bit long and repetitive in some places and hence could be more condensed and streamlined). Hence, I suggest this paper can be accepted with minor revision. Please see my specific comments below.

Some minor comments:

Abstract:

Line 23: *Yet, the timing and molecular consequences of the microbiome during the 24 strobilation process had not been investigated.* Is strangely formulated – *the timing and molecular consequences of the microbiome*- please re-write.

Introduction:

Line 86-87: *Several proteins, including transcriptional regulators have been shown to be involved in the induction of strobilation.* Reference missing.

Line 97-98: *These expression of these four genes is strongly upregulated in polyps entering the strobilation phase. CL112 97 encodes a secreted protein containing an epidermal growth factor-like domain and be expressed in endodermal cells of each segment of the early strobili.* Reference missing

Results/Methods:

General comment: can you please justify use of V1-V2 region? Some microbiota could be missed in this case. Did you try other primer sets and compare?

Methods:

Line 406:

For isolation of A. aurita-associated microbiota, 10 polyps, 10 early strobilae (5 days 407 post-induction), and 10 late strobilae (9 days post-induction) were transferred into 408 500 µl sterile ASW and washed three times to remove transient bacteria. The animals were collectively homogenized with a motorized pestle (KONTES, DWK Life Science, Wertheim, Germany) and filtered through a 3.1 µm filter (Lab Logistic group, Meckenheim, Germany) to remove eukaryotic cells. This is not clear to me. Did you combine/pool microbiota from all this different 'hosts' to obtain inoculum or you kept it separately for each 'host'? Depending on what exactly you did in my opinion affects the outcome substantially.

Line 412: *Aliquots (105 cells/ml) of the filtrate were immediately frozen in liquid nitrogen and stored at -80 °C in the presence of 10 % glycerol. These frozen stocks served as inoculum for all recolonization experiments presented here.* I wonder how many of cells lost their viability via this process – maybe also specific taxa...did you maybe check the effect on microbial community?

Line 482: *Recolonization of sterile polyps, early and late strobila was conducted by addition of 10 µl of the thawed inoculum to ASW (105 cells/ml and per animal, see above), with fresh ASW containing inoculum applied the next day, so that the animals were exposed to the inoculum for 48 h. A control of inoculum was incubated for 48 h in ASW only.* At this point I started to think about viable/dead cells within inoculum and then I realized that you used STYo0 to count – which allows you to distinguish live/dead cells, however maybe not all readers know that so would be nice to specify this somewhere..else makes you wonder how much viability of cells was affected by your freezing/thawing of inoculum, hence also affecting your community composition,

Results:

Line 149: what about alpha diversity of whole community?

Line 152 (and other cases): when you talk about relative abundance one needs to be aware that of course this depends on total cell abundance, e.g., one taxa can increase in relative abundance, but overall cell abundance (cells/mL) is lower and this is important!, also an increase/decrease can be due to decrease/increase of other taxa...it can give a wrong impression! Since I understood you used SYTO9 to distinguish between live and dead and made an effort to count cells on polyps etc., why not making an effort to take this actual cell numbers into account

Line 156-157: sentence not clear, please re-write: *Overall, the microbiota of recolonized polyps more strongly resembled that of native polyps than bacteria maintained in artificial seawater (ASW) in absence of animals.*

Line 216-216: *The individual administration of antibiotics also reduced the native bacterial community monitored by a decrease of 30 - 80 % in CFU/ml.* This is confusing to me: when you talk about CFU/mL I assume only cultivable part of community was assessed? But to my understanding this was not the case. Please explain.

Line 270-272: *For this, native polyps with their diverse, healthy microbiome were incubated together with sterile polyps, but separated by a 0.2 µm filter that allowed the transfer of metabolites, but not of bacteria.* If we are picky we would take into account the fact that some really small bacteria can escape 0.2 µm filtration!

Discussion:

Line 287: *Our results not only illustrate how important a native microbiome is for the asexual reproduction of A. aurita, but also demonstrated that the generation of offspring is only possible when the microbiome is present before the animal enters the strobilation process.* Just a general thought: I cannot image where and when in natural scenario microbiome would not be present before animal enters strobilation process, so not really sure, from ecological perspective, why this is even relevant. Maybe discussion on this aspect could be elaborate.

Line 332-334: *The abundance of Vibrio and Colwellia is increasing in recolonized polyps, whereas ASW shifted the bacterial population to a significantly structure accompanied with a lower α -diversity.* This is a very strange sentence, I don't understand what is meant here. Please correct.

Line 336-337: *During a reassembly of the microbiota on the host's mucus, interactions between host and bacteria as well as amongst bacteria ensure a specific microbial community pattern of Aurelia.* How can you possibly know this? Did you test other jellyfish species and compare species-specificity of associated microbiota?

**Title**

**The presence of a microbiome in *Aurelia aurita* polyps is required for the**
**polyp-to-jellyfish transition prior to the initiation of the strobilation process**

Running title

The microbiome in pre-strobilation polyps ensures the *A. aurita* life cycle

Nadin Jensen¹, Nancy Weiland-Bräuer¹, Shindhuja Joel¹, Cynthia Maria Chibani¹,
Ruth Anne Schmitz^{1*}

¹Institute of General Microbiology, Christian-Albrechts University Kiel, Am
Botanischen Garten 1-9, 24118, Kiel, Germany

*rschmitz@ifam.uni-kiel.de

Keywords: , recolonization, life cycle

Word counts abstract: 350 (200 abstract, 148 importance)

Word counts text: 5555 (Materials/Methods section included)

**Abstract**

*Aurelia aurita*'s intricate life cycle alternates between benthic polyp and pelagic
medusa stages. The strobilation process, a critical asexual reproduction mechanism
in this jellyfish, is severely compromised in the absence of the natural polyp
microbiome, with limited production and release of ephyrae. Yet, the recolonization of
sterile polyps with a polyp-native microbiome can correct this defect. Here we
investigated the precise timing necessary for recolonization as well as the host-
associated molecular processes involved. We deciphered that a natural microbiota
had to be present in polyps prior to the onset of strobilation in order to ensure normal
asexual reproduction and a successful polyp-to-jellyfish transition, meaning that
adding microbiota to sterile polyps after the onset of strobilation failed to restore the
normal strobilation process. The absence of a microbiome was associated with
decreased transcription of developmental and strobilation genes; these were
exclusively observed for native polyps and sterile polyps that were recolonized before
the initiation of strobilation. Direct cell contact between the host and its associated
bacteria was also required for the normal production of offspring. Overall, our findings
indicate that the presence of a native microbiome at the polyp stage prior to the
onset of strobilation is essential for a normal polyp-to-jellyfish transition.

**Importance**

All multicellular organisms are associated with microorganisms that play a
fundamental role in the health and fitness of the host. Notably, the native microbiome
of the Cnidarian *Aurelia aurita* is crucial for the asexual reproduction of the jellyfish.
Sterile polyps display malformed strobilae and a halt of ephyrae release that can be
restored by recolonizing sterile polyps with a native microbiota. Despite that, little is

known about the strobilation process's timing, molecular consequences, and
microbial impact. The present study shows that the life cycle of *A. aurita* depends on
the presence of the native microbiome at the polyp stage prior to the onset of
strobilation to ensure the polyp-to-jellyfish transition. Moreover, sterile individuals
correlate with reduced transcription levels of developmental and strobilation genes,
evidencing the microbiome's impact on strobilation on the molecular level.
Transcription of strobilation genes was exclusively detected in native polyps and
those recolonized before initiating strobilation, suggesting a microbiota-dependent
gene regulation.

**1 Introduction**

[revised manuscript text omitted]

We recently described how in the absence of a native microbiota, the fitness of *A.*
*aurita* is severely impaired, and its asexual reproduction is halted (11). Recolonizing
sterile polyps with native polyp-derived microbiota restored asexual development
and reproduction. However, whether the developmental restoration of a sterile polyp
by recolonization is possible at multiple developmental stages, or whether the precise
timing of the host-microbiota interaction is important remains to be determined.
Moreover, the underlying molecular pathways are unknown.

Here, we aimed to identify the developmental stage(s) apt for the recolonization of
sterile animals in order to restore asexual reproduction, and to gain insights into the
role of the microbiota in this process. Sterile animals were recolonized with native

microbiota at three different life stages: polyp prior to the initiation of strobilation,
early strobila and late strobila. The formation of strobilae and the release of ephyrae
was evaluated and phenotypic effects were recorded. The gene expression of
essential developmental- and strobilation genes was examined to determine the
effect of the exact timing of recolonization for asexual reproduction, focussing on the
molecular level.

**2 Results**

The life cycle of *A. aurita* is summarized in Fig. 1A. Sexual reproduction of the mature
medusa results in the release of planula larvae that develop into benthic polyps.
Asexual reproduction can be induced chemically to start the transition into early and
then late strobila, forming bodily segments to release precursor medusa in the form
of ephyra. This offspring generation is impaired in the absence of a microbiota, but
can be restored by recolonization (11). The experimental setup to study the effect of
different timing of this recolonization is summarized in Fig. 1B. As a first step, sterile
polyps were recolonized (treatment R1) and the associated microbiota was
characterized.

**Recolonization of sterile polyps with a native microbiota demonstrates a strong** 137 **host impact on the microbial community structure**

The microbiota used as inoculum to recolonize sterile polyps was isolated from native
polyps and characterized by amplicon sequencing of the V1-V2 region of the 16S
rRNA gene. Following recolonization of sterile polyps, their microbiota was
characterized again and compared to that of truly native polyps (Fig. 2). In all three
microbiota samples, the classes of Alpha- and Gammaproteobacteria dominated. In

the native polyps, unclassified representatives of Gammaproteobacteria were
dominant, followed by the genera *Pelagibus* and *Vibrio* (Fig. 2A, first column). The
major representatives in the inoculum were (in descending order) *Vibrio*,
*Terasakiella*, *Colwellia*, *Flavobacterium* and *Sphingorhabdus*. The diversity of Alpha-
and Gammaproteobacteria was similar in native polyps and inoculum, resulting in no
significant differences in the α -diversity (Fig. 2B). Thus, although differences in
abundances were noted, the inoculum was considered similar to the microbiota of
native polyps. Recolonization of sterile polyps with the inoculum shifted the bacterial
community to an increased relative abundance of *Colwellia* and *Ahrensia*. Compared
to the native polyps, recolonized polyps had larger fractions of *Vibrio*, *Ahrensia*,
*Cohaesibacter*, and *Flavobacterium* (Fig. 2A). This is in agreement with a significant
increase in α -diversity (Fig. 2B). Overall, the microbiota of recolonized polyps more
strongly resembled that of native polyps than bacteria maintained in artificial
seawater (ASW) in absence of animals. The large difference in the community
structure of the latter was reflected by a significantly lower α -diversity, indicating a
strong impact of the host on the microbial community structure. In summary,
recolonization of the sterile polyp with an inoculum derived from native microbiota
resulted in a microbial community pattern similar to those observed in native polyps.

**The presence of a microbiota affects the transcription of host developmental** 162 **genes**

In order to uncover whether and how the microbial presence influences life cycle
decisions of *A. aurita* on the molecular level, transcription levels of the four *A. aurita*-
specific genes shown to be involved in strobilation (*CL112*, *CL355*, *CL390*, and
*CL631*) were compared between native and sterile animals (Fig. 3A, left panel) [33,
42]. Since the life cycle of sterile animals is disturbed, the latter were analyzed at the

time points where the normal developmental life stages could be assumed, although
phenotypically the animals were abnormal. In native polyps, none of the investigated
genes were expressed, giving negative ΔCt values after normalization (Fig. 3A).
Once strobilation had started, their transcription levels rose to detectable levels
($\text{Ct}_{\text{CL112}} = 26.6$; $\text{Ct}_{\text{CL355}} = 22.9$; $\text{Ct}_{\text{CL631}} = 21.9$; $\text{Ct}_{\text{CL390}} = 18.4$). Their maximal
expression was reached during the formation of late strobilae ($\text{Ct}_{\text{CL112}} = 22.6$; Ct_{CL355}
$= 18.9$; $\text{Ct}_{\text{CL631}} = 16.7$; $\text{Ct}_{\text{CL390}} = 15.9$) and decreased again during the ephyra stage
(Fig. 3A, left panel). Under sterile conditions, the *A. aurita*-specific strobilation genes
were not expressed at all (Fig. 3A, right panel, for Ct values see Suppl. Table S1).

The gene expression analysis was extended with the developmental genes *DNMT*,
*RxR*, *BMP*, *Wnt1*, and *Ptc*. These genes produced similar expression patterns over
time during the development of native animals, but were not expressed under sterile
conditions (Fig S1).

The differential gene expression (fold changes) between the native and respective
sterile life stages is visualized in Fig. 3B, for the strobilation genes as well as the
conserved developmental genes. An overall down-regulation was observed in sterile
animals, except for the polyp stage where none of these genes is expressed. The
down-regulation of the specific strobilation genes was notable, up to a factor of
4×10^6 , while the general developmental genes decreased more moderately. The
transcription of all four strobilation genes was most strongly decreased in sterile late
strobilae, while *CL355* and *CL390* transcription levels were also strongly decreased
in sterile early strobilae (10^5 -fold decrease). At the ephyra stage of sterile animals,
the transcript level of *CL355* was still 10^6 -fold lower than that of native ephyra.
Together, these data show reduced or absent transcription of 5 developmental and 4

strobilation-associated genes during *A. aurita* strobilation in the absence of a
microbiota

**Timing of the microbial presence is crucial for life cycle of *A. aurita***

[revised manuscript text omitted]

ephyrae release by sterile polyps, whereas native polyps showed normal offspring
output (Fig. 6). From this, we conclude that a direct contact between host and
bacteria is needed and only that leads to a healthy strobilation process. During this
direct contact a host receptor might directly perceive a signal, either due to the
contact itself or via a bacterial product that is produced in response to the contact.

**3 Discussion**

Our findings illustrate not only the importance of a native microbiome for the
strobilation process in *A. aurita* but also that normal offspring generation is only
possible when the microbiome is present before the animal enters strobilation.

The absence of microbiota and the resulting phenotypic and developmental
consequences for the animal correlate with the regulation of developmental and
strobilation genes (Fig. 3). Recolonization experiments on sterile animals only
resulted in native expression patterns of strobilation genes, with full segmentation of
the polyp and healthy offspring generation, when performed before the polyp entered
the molecular strobilation process. Recolonization during the ongoing strobilation
process (early and late strobila) neither molecularly nor morphologically restored this
part of the life cycle. Instead, the treatment resulted in lower transcription levels, as
seen in sterile polyps, as well as a malformation of the reproduction phenotype. We
cannot be sure that the morphological and molecular observations of sterility are
exclusively based on the absence of the native microbiome. Off-target antibiotic
effects may also result in deformed *A. aurita* polyps, as has been reported for single
antibiotics in various animals (46, 47). Nonetheless, those authors did not experiment
under the premises of the metaorganism/holobiont concept and thus viewed the
consequences of antibiotics without considering the microbiome as a key player, in
that the effects of impairments on the animal were caused by antibiotic-induced

microbiome disturbances. Since fitness traits related to strobilation can be restored
after recolonization of sterile *A. aurita* polyps with the native microbiota, it is likely that
the observed sterility is rather related to the microbiome than to the toxicity of the
antibiotic, implying that antibiotics are the cause, but not the reason for strobilation
malfunctioning in polyps. In the absence of their respective microbiome, the
cockroach *Periplaneta americana* exhibits a starvation-like transcriptional response in
growth and development (48). Also, the fruit fly *Drosophila* requires its associated
microbiome for reproduction, and the microbial composition and timing of this
microbial presence have a significant impact on the lifespan of the insect (12).
Recolonization of germ-free *Drosophila* embryos restored their development (49, 50),
as it did with *A. aurita*.

In mice, the gene expression of mucus regulatory proteins differs between native and
germ-free animals, indicating that specific groups of bacteria accelerate the onset of
development during postnatal development (51). The impact of a natural microbiota
on the development of animal hosts has been observed across phyla, and the timing
of its presence, which most likely includes microbe-host interactions, is clearly
essential for a natural development.

*Aurelia* polyps harbor a specific bacterial community with a high diversity that
undergoes significant restructuring during the polyp-to-jellyfish transition (28, 52). The
specific microbial community is assumed to be essential for the sessile life stage
(28). In return, the bacteria likely benefit from the polyp's mucus as a source of
nutrients (28, 53, 54). We observed that polyps with naturally associated native
microbiota and sterile polyps after recolonization harbored more similar microbial
communities in terms of diversity than either of them with the bacterial population in
the surrounding artificial seawater (Fig. 2), which corroborates with many other

studies and points to an impact of the host on the establishment and maintenance of
the microbiota, resulting in the same molecular and phenotypic consequences.
During a reassembly of the microbiota on the host's mucus, interactions between
host and bacteria as well as amongst bacteria ensure a specific microbial community
pattern of *Aurelia*. Bacteria-bacteria interactions are assumed to shape the
microbiome of *Nematostella* dynamically according to the host's developmental stage
(56, 57). Such interactions can include secreted bacterial compounds that kill or
impair their neighbours (56, 58, 59). Examples of host-bacteria interactions shaping
the microbial community composition also exist (60, 61). Distinct host-bacteria
interactions are widely based on host receptors exchanging information (2). Such
information affects or promotes the susceptibility to colonize a host by different
molecular mechanisms, often involving pattern recognition receptors or Toll-like
receptors that can recognize a wide range of microbe-derived signals (2, 62).

In our experiments, certain bacteria were missing in the microbiota of recolonized
polyps compared to native polyps, indicating redundant microbiome members, or at
least members not required for normal strobilation. Interestingly, recolonization
increased the abundance of *Vibrio* compared to native polyps (Fig. 2A). *Vibrionaceae*
were also found with a high abundance in corals and in the mucus of other marine
invertebrates (63, 64). Overall, *Vibrionaceae* display only slight host preference by
efficient dispersal-colonization dynamics mediated by preferred food sources (65). In
*Aurelia* medusae, *Vibrionaceae* dominated the bacterial community, and this led to
speculations on different roles of *Vibrio* during the different life stages of the jellyfish
(53). Nonetheless, the abundance of *Vibrio* we reported may only be a snapshot
representing the situation after two days of recolonization.

The pathways of microbes modulating the host's metamorphosis or developmental
processes have been reported for various animals (66-68). In *A. aurita*, we
demonstrate that a microbial presence is essential before the animals enter the
strobilation process and that it affects the transcription of strobilation genes. This
could be related to microbe-derived molecules and/or a direct host-microbe contact.
Several studies have demonstrated that a bacterial metabolite or product initiates
host metamorphosis (69-71). *Pseudoalteromonas luteoviolacea* requires physical
contact with *Hydroides elegans* to produce syringe-like structures, called
metamorphosis-associated contractile structures (MACs) that inject stimulatory
proteins crucial for the metamorphosis of the worm (72). Soluble bacteria-derived
products of an as yet unknown nature were shown to induce settlement and
metamorphosis of oyster larvae (73). To determine whether direct contact between
*Aurelia* and bacteria is required for strobilation, an experimental design with a
membrane that separated sterile and native polyps was used. This ensured the
absence of direct host-bacteria contact, but allowed the potential transfer of
molecules and enzymes to the sterile polyp (Fig. 6). Since no asexual reproduction in
the latter was obtained, we assume that direct contact between host and bacteria is
required and only this leads to a healthy strobilation process in *A. aurita*. It remains to
be elucidated whether the polyp directly perceives the signal due to the physical
contact itself, or a bacterial signal is transferred to the host in response to such
contact.

In conclusion, our results identified the gene regulation of strobilation associated
genes that would not occur in the absence of a polyp-associated microbiome prior to
entering the strobilation process. Microbe presence, and possibly interaction play a
key role in the development of this metazoan. The presented study suggests that a

well-timed direct contact between the polyp and its associated bacteria enables the
induction of a healthy metamorphosis, and co-occurs with the presence of
transcribed host genes. Whether the direct contact enables putative bacterial
molecules to be transferred to the host, or alternatively receptors that may be
activated by physical contact needs to be investigated in future experiments. Also,
using reduced manipulated communities might decipher specific bacteria responsible
and/or allow for the identification of interaction-related molecules, if any. The
identification of such molecules may provide insights into the microbial contribution
that allow the polyp-to jellyfish transition in this evolutionary ancient metazoan.

**4 Methods**

**Husbandry of *Aurelia aurita***

*Aurelia aurita* polyps were derived from the North Atlantic (Roscoff, France) almost
15 years ago and have since been kept under standard husbandry conditions in
artificial seawater (ASW) with a salinity of 30 practical salinity units at 19 °C as
described in Weiland-Bräuer *et al.* (11, 74, 75). The polyps were fed twice a week
with freshly hatched *Artemia salina* (HOBBY, Grafenschaft-Gelsdorf, Germany), and the
water was changed once a week.

**Isolation and characterization of *A. aurita* microbiota**

For isolation of *A. aurita*-associated microbiota, 10 polyps before the initiation of
strobilation, 10 early strobilae (5 days post-induction), and 10 late strobilae (9 days
post-induction) were transferred into 500 µl sterile ASW and washed three times to
remove transient bacteria. The animals were collectively homogenized with a
motorized pestle (KONTES, DWK Life Science, Wertheim, Germany) and filtered

through a 3.1 µm filter (Lab Logistic group, Meckenheim, Germany) to remove
eukaryotic cells. Aliquots (10^5 cells/ml) of the filtrate were immediately frozen in liquid
nitrogen and stored at -80 °C in 10 % glycerol. These frozen stocks served as
inoculum for all recolonization experiments. Cell numbers were determined by
fluorescence microscopy (Axio Scope microscope and Axio Vision software; Zeiss,
Jena, Germany) using SYTO9 (1:1000 (Invitrogen, Darmstadt, Germany)) and a
Neubauer count chamber (Assistant, Sondheim vor der Röhn, Germany) according to
the manufacturer's instructions. The absolute number of cells per polyp was
calculated as 2.8×10^6 cells for one single polyp.

Total bacterial DNA was extracted from thawed inoculum using the Wizard Genomic
DNA Purification kit (Promega) and the V1 and V2 region of the 16S rRNA gene were
amplified as described in Weiland-Bräuer *et al.* (11) using 27 F/338 R primers (27 F:
5'-AGAGTTTGATCCTGGCTCAG-3', 338 R: 5'-TGCTGCCTCCCGTAGGAGT-3'). All
primers contained unique identifier sequences (barcodes) to distinguish between the
samples following published methods (76) with an optimized primer design (77).
Amplification was performed in 20 µl using Phusion high-fidelity DNA polymerase
(New England Biolabs) with a protocol of 98°C, 30 s followed by 30 cycles (98°C, 9 s;
55°C, 60 s; 72°C, 90 s), 10 min extension at 72°C, and 10 min 10°C cooling. The
same procedure was used to characterize the microbiota of six replicates of
homogenized polyps or recolonized animals (see below).

All amplicons were purified using the MiniElute gel extraction kit (Qiagen, Hilden,
Germany), quantified using the Quant-iT PicoGreen kit (Invitrogen, Darmstadt,
Germany) and sequenced on an Illumina platform, for which equal amounts were
pooled.

Paired-end raw read files were processed using QIIME 2 (version qiime2-2021.2
(78)). The 16S rDNA sequences were denoised using DADA2 via q2-dada2. Quality
profile plots were inspected to select values for truncation of forward and reverse
reads. The truncation and trimming were set to "--p-trim-left-f 13, --p-trim-left-r 13;
and --p-trunc-len -f 250,--p-trunc-len-r 250". All resulting amplicon sequence variants
(ASVs) were aligned with mafft via q2- and this was used to construct a mid-point
rooted phylogeny with fasttree2. Bacterial taxonomic assignment was done using
feature-classifier classify-consensus-vsearch, against the Silva 132 v. 99 QIIME2-
compatible database. Alpha-diversity metrics based on observed ASVs were
estimated using q2-diversity after samples were rarefied (subsampling without
replacement) to 2.106 sequences per sample to determine differences among the
bacterial communities driven by different treatments. Sequence data for all replicates
and treatments were deposited under the NCBI BioProject [PRJNA896887](https://www.ncbi.nlm.nih.gov/bioproject/PRJNA896887) comprising
locus tag prefixes SAMN31571268 to SAMN31571288.

**Generation of germ-free polyps**

Preliminary experiments with single and mixed antibiotics (chloramphenicol,
neomycin, ampicillin, streptomycin, rifampicin (each 50mg/L) and 60 mg/L
spectinomycin) were performed to establish the final protocol for the generation of
sterile polyps. The remaining cultivable bacteria were analyzed by CFU/ml on MB
and R2A agar plates after 5 days of incubation. Consequently, sterile polyps were
generated by treatment of 3-day starved animals with an antibiotic mixture containing
50 mg/L each of chloramphenicol, neomycin, ampicillin, streptomycin, rifampicin and
60 mg/L spectinomycin (all from Carl Roth, Karlsruhe, Germany) in sterile ASW. After
5 days, absence of cultivable bacteria was ensured by plating polyp homogenate on

MB and R2A agar plates. In addition, the genomic DNA of the polyps was used as
template for full-length 16S rRNA gene amplification using GoTaq polymerase with
universal primers 27F (5`-AGAGTTTGATCCTGGCTCAG-3`) and 1492R (5`-
GGTTACCTTGTACGACTT-3`). Absence of amplicons in combination with absence
of colonies confirmed the polyps were sterile.

**Recolonization of sterile *A. aurita* at different developmental stages**

Single sterile polyps in individual wells of a 48-well plate containing 1 ml sterile ASW
were recolonized at three developmental stages with 96 replicates per treatment.
Strobilation was induced with 5 µM 5-methoxy-2-methyl indole (Carl Roth, Karlsruhe,
Germany) for three days. From day 4, the inducer was omitted (unless stated
otherwise) and developing segments and the release of ephyrae was recorded daily
using a stereomicroscope (Novex Binoculares RZB-PL Zoom-Microscope 65.500,
Novex, Arnhem, the Netherlands) equipped with a high-definition multimedia
interface (HDMI)/HD camera. The stage of early strobilation was determined by the
appearance of the first strobila , as detected by the constriction of the first segments
at the apical part, and early strobilation was typically complete on day 5 under native
conditions (NC). The stage of late strobilation was reached when segmentation was
complete (at day 9 for NC), while the sequential release of ephyrae was observed
from day 12 onwards. The release of ephyrae was monitored until the end of the
experiment on day 23.

Recolonization of sterile polyps, early and late strobila was conducted by addition of
10 µl of the thawed inoculum to ASW (10^5 cells/ml and per animal, see above), with
fresh ASW containing inoculum applied the next day, so that the animals were
exposed to the inoculum for 48 h. A control of inoculum was incubated for 48 h in

ASW only. Treatment group R1 (sterile polyps) were recolonized prior to induction.
Treatment group R2 was recolonized at **the early strobila stage** and treatment group
R3 at **late strobila stage**. Native animals (NC) and a sterile control (SC) were
included. Treatments R2 and R3 were also performed in constant presence of the
inducer, giving R2i and R3i, respectively, to assess the lasting effect of the inducer.
To consider the possibility that a life stage-specific microbiota is required for normal
progeny output recolonization of sterile early strobila (R2esm) and late strobilae
(R3lsm) was conducted by adding 10 µl of the thawed inoculum of the respective life
stages to ASW (10^5 cells/ml and per animal, see above).

Preliminary experiments with eight replicates for identifying a metabolite of bacterial
origin or the metabolization of a host molecule by bacteria without direct contact were
conducted with polyps in 1 mL sterile ASW in single 48 well microtiter plate cavities.
A single sterile polyp was settled at the bottom of each cavity. A 0.2 µm centrifugal
sterile filter unit (amchro, Hattersheim, Germany) was inserted into the cavity and 10
native polyps were added to the unit. Strobilation was induced after 8 days post-
adding of native polyps using 5 µM 5-methoxy-2-methyl indole (this inducer can pass
the filter). Segmentation and ephyrae release were recorded for 20 days.

**Expression analysis of developmental genes by qRT PCR**

Expression analysis of host genes was performed with 6 replicates of four
development stages (polyp, early strobila, late strobila, and ephyra) for the 5
treatments (NC, SC, R1, R2, and R3). A total of 9 genes were assessed (Table 1).
For RNA extraction, animals were transferred to 1.5 ml reaction tubes, washed three
486 times with sterile ASW, frozen in liquid nitrogen and stored at -80 °C until use.
Thawed animals were homogenized and total RNA was isolated by phenol-
chloroform extraction as follows: 200 µl of 100 mM Tris/HCl (pH 5.5), 10 mM EDTA,

0.1 M NaCl, 1 % SDS, 1 % 2-mercaptoethanol and 2 μ l Proteinase K (25 mg/ml;
Thermo Fisher Scientific, Darmstadt, Germany) was added to the homogenate and
incubated for 10 min at 55 °C. Following, protein precipitation with 5 μ l 3M potassium
acetate, 250 μ l ROTI® Aqua-P/C/I (Carl Roth, Karlsruhe, Germany) for 15 min on ice,
and centrifugation (14,000 rpm for 15 min, 4 °C) the RNA was precipitated by adding
1 volume of 2-propanol. RNA pellets were washed once with 70 % EtOH and finally
dissolved in 30 μ l RNase-free water (Carl Roth, Karlsruhe, Germany).

Prior to reverse transcription, DNA was removed by DNase I treatment (Thermo
Fisher Scientific, Darmstadt, Germany). Successful DNA removal was ensured by
standard PCR analysis using the GoTaq PCR kit (Promega, Madison, WI, USA) with
primers 27F and 1492R. RNA was reverse transcribed using Random Hexamers of
SuperScript™ IV First-Strand Synthesis System (Invitrogen). cDNA was purified
using the PCR Clean-Up Kit (Macherey-Nagel, Düren, Germany).

The transcripts of 10 selected target genes were analysed by qPCR in 25 μ l
Platinum™ SYBR™ Green qPCR SuperMix-UDG (Invitrogen) with 5 μ l cDNA (20 ng)
and 0.4 μ M forward and reverse primers, as listed in Table 1. The cycling conditions
were 50 °C, 2 min followed by 95 °C, 2 min and 40 cycles of 95 °C, 15 s; 60 °C, 30 s;
95 °C, 15 s and then 60 °C, 30 s and 95 °C, 15 s. Average cycle threshold (Ct)

[revised manuscript text omitted]

65. Preheim SP, Boucher Y, Wildschutte H, David LA, Veneziano D, Alm EJ, Polz MF. 2011. Metapopulation structure of *Vibrionaceae* among coastal marine invertebrates. *Environ Microbiol* 13:265-275.

66. Shikuma NJ. 2021. Bacteria-Stimulated Metamorphosis: an Ocean of Insights from Investigating a Transient Host-Microbe Interaction. *mSystems* doi:10.1128/mSystems.00754-21:e0075421.
67. Cavalcanti GS, Alker AT, Delherbe N, Malter KE, Shikuma NJ. 2020. The Influence of Bacteria on Animal Metamorphosis. *Annu Rev Microbiol* 74:137-158.
68. Guo H, Rischer M, Westermann M, Beemelmans C. 2021. Two Distinct Bacterial Biofilm Components Trigger Metamorphosis in the Colonial Hydrozoan *Hydractinia echinata*. *mBio* 12:e0040121.
69. Sharp KH, Sneed JM, Ritchie KB, McDaniel L, Paul VJ. 2015. Induction of Larval Settlement in the Reef Coral *Porites astreoides* by a Cultivated Marine *Roseobacter* Strain. *Biol Bull* 228:98-107.
70. Yang F, Mo J, Wei Z, Long L. 2020. Calcified macroalgae and their bacterial community in relation to larval settlement and metamorphosis of reef-building coral *Pocillopora damicornis*. *FEMS Microbiol Ecol* 97.
71. Webster NS, Smith LD, Heyward AJ, Watts JE, Webb RI, Blackall LL, Negri AP. 2004. Metamorphosis of a scleractinian coral in response to microbial biofilms. *Appl Environ Microbiol* 70:1213-21.
72. Shikuma NJ, Pilhofer M, Weiss GL, Hadfield MG, Jensen GJ, Newman DK. 2014. Marine tubeworm metamorphosis induced by arrays of bacterial phage tail-like structures. *Science* 343:529-33.
73. Fitt WK, Coon SL, Walch M, Weiner RM, Colwell RR, Bonar DB. 1990. Settlement behavior and metamorphosis of oyster larvae (*Crassostrea gigas*) in response to bacterial supernatants. *Marine Biology* 106:389-394.
74. Dawson MN, Jacobs DK. 2001. Molecular Evidence for Cryptic Species of *Aurelia aurita* (Cnidaria, Scyphozoa). *The Biological Bulletin* 200:92-96.
75. Schroth W, Jarms G, Streit B, Schierwater B. 2002. Speciation and phylogeography in the cosmopolitan marine moon jelly, *Aurelia* sp. *BMC Evolutionary Biology* 2:1.
76. Kozich JJ, Westcott SL, Baxter NT, Highlander SK, Schloss PD. 2013. Development of a dual-index sequencing strategy and curation pipeline for analyzing amplicon sequence data on the MiSeq Illumina sequencing platform. *Appl Environ Microbiol* 79:5112-20.
77. Fadrosch DW, Ma B, Gajer P, Sengamalay N, Ott S, Brotman RM, Ravel J. 2014. An improved dual-indexing approach for multiplexed 16S rRNA gene sequencing on the Illumina MiSeq platform. *Microbiome* 2:6.
78. Bolyen E, Rideout JR, Dillon MR, Bokulich NA, Abnet CC, Al-Ghalith GA, Alexander H, Alm EJ, Arumugam M, Asnicar F, Bai Y, Bisanz JE, Bittinger K, Brejnrod A, Brislawn CJ, Brown CT, Callahan BJ, Caraballo-Rodríguez AM, Chase J, Cope EK, Da Silva R, Diener C, Dorrestein PC, Douglas GM, Durall DM, Duvallet C, Edwardson CF, Ernst M, Estaki M, Fouquier J, Gauglitz JM, Gibbons SM, Gibson DL, Gonzalez A, Gorlick K, Guo J, Hillmann B, Holmes S, Holste H, Huttenhower C, Huttley GA, Janssen S, Jarmusch AK, Jiang L, Kaehler BD, Kang KB, Keefe CR, Keim P, Kelley ST, Knights D, et al. 2019. Reproducible, interactive, scalable and extensible microbiome data science using QIIME 2. *Nat Biotechnol* 37:852-857.
79. Livak KJ, Schmittgen TD. 2001. Analysis of Relative Gene Expression Data Using Real-Time Quantitative PCR and the 2- $\Delta\Delta$ CT Method. *Methods* 25:402-408.

Author contributions

N.W.-B. and R.A.S. designed the research; N.J. and performed the research with support from S.J.; C.M.C. analyzed the sequence data; N.J., N.W.-B. and R.A.S. wrote the paper; N.W.-B. and R.A.S. edited the paper.

Competing Interests Statement

The authors declare no competing interests.

Figure legends

Figure 1. Life cycle of *Aurelia aurita* and study design for recolonization at various developmental stages. (A) Summary of the life cycle of the moon jellyfish *Aurelia aurita*. Sexual reproduction of the matured medusa results in the release of planula larvae that develop into benthic polyps. Asexual reproduction includes the transition into early and then late strobila, forming bodily segments to release precursor medusa in the form of ephyra. This offspring generation is impaired in the absence of a microbiota. (B) Experimental design to identify permissible time point(s) of recolonization (R1, R2, R3) and in constant presence (R2i, R3i) of inducer.

Figure 2. Analysis of the microbial community of *Aurelia aurita*. (A) Amplicon sequencing variants (ASVs) abundance based on the V1 – V2 region of the 16S rRNA gene of the bacterial communities derived from native polyps, the inoculum used for recolonization, obtained microbiota from recolonized sterile polyps and a control of the inoculum incubated in artificial seawater (ASW) is presented at the genus level and normalized by the total number of reads per sample. Genera with a relative high abundance in at least one of the four samples are shown in bold font. Bar plots represent averages of six replicates per treatment. Taxa with a mean

relative abundance < 1 % across all samples are collectively reported as "Others". In the electronic version a scroll over the colored box and a right mouse click will show the genus. **(B)** Box plot of Alpha diversity (Shannon index). Significance of comparisons is shown as * for P-value <0.05 and ** for P-value < 0.01.

Figure 3. Transcription of *Aurelia*-specific strobilation genes depends on presence of a microbiota. (A) Transcription patterns (reported as Δ Ct using *EF1* for normalization) of *Aurelia*-specific strobilation genes *CL112*, *CL355*, *CL390*, and *CL631* in the life stages of polyps, early strobila, late strobila, and ephyra, of native (left) and sterile (right) animals. Averages of four biological replicates with three technical replicates each are reported. **(B)** Fold changes of transcription of selected conserved developmental genes and *A. aurita*-specific strobilation genes in sterile polyps compared to the corresponding native life stage.

Figure 4. Timing of recolonization is crucial to restore asexual reproduction. (A) Morphology of polyps undergoing strobilation under various conditions, with left to right: native conditions (NC), sterile conditions (SC), and recolonization of sterile polyps (R1), of sterile early strobila (R2), of sterile late strobila (R3). Photographs show the phenotypical appearance of, from the top, polyps, early strobilae, late strobilae and ephyrae. The number of days post infection is specified. Scale bars correspond to 1 mm. **(B)** Bar plot summarizing the number of segments per animal at late strobila stage (9 days post induction) and of released ephyrae per animal (14 days post-induction). Significance of comparisons are shown as **** for P-value < 0.0001. **(C)** Phenotypes of sterile polyps undergoing strobilation when sterile early strobilae (R2esm) and sterile late strobilae (R3lsm) were recolonized with their life stage-specific microbiota.

Figure 5. Impact of the native microbiota on the transcription of *A. aurita*-specific strobilation genes during asexual reproduction. (A) Heat map of normalized Ct values (Δ Ct) of genes *CL112*, *CL355*, *CL390*, and *CL631* using the housekeeping gene *EF1* for normalization. Δ Ct values represent an average of four biological replicates with three technical replicates for each. (B) Expression profile of *CL390* during development over time for the different treatments. A complete summary of all assessed genes is presented in Fig. S3.

Figure 6. Direct contact between polyp and bacteria is assumed for a normal strobilation process. Strobilation was induced with the chemical inducer that is able to pass the membrane (0.2 μ m sterile filter). (A) A single native polyp was physically separated by the membrane from the strobilation inducer – ASW solution. (B) A single sterile polyp was separated by the membrane that hinders microbial cells to pass from a single or a pool of ten pool polyps. (C) A single sterile polyp was separated by the membrane that hinders microbial cells to pass from a single or a pool of ten pool polyps. After eight days of pre-incubation, strobilation as induced.

Table

Table 1: Primers used for qRT PCR analysis of 10 target genes

Gene	Primer Name	Sequence (5'-3')	Length (bp)	Tm (°C)	Gene function	Reference
EF1	EF1_F2	AGTTCCAGGGGACAATGTTGG	21	59.8	Protein biosynthesis	(33)
	EF1_R2	TGGATTTCCGCCAGGATGGTTC	21	59.8		
CL112	aCL112_F2	GAAGCTACCAGATCCGTTTG G	21	59.8	Induction of strobilation	(33)
	aCL112_R2	TGCAAGCGCATCTGTTCACAG	21	59.8		
CL355	CL355_for	TTCCGGAGAGCAGACCAATG	20	59.4	Induction of strobilation	(42)
	CL355_rev	CCAACGGCTGCATATACCATC	21	59.8		
CL390	CL390_F2	AAGGTGCGACAATGAAGGTCC	21	59.8	Induction of strobilation	(33)
	CL390_R2	GTCTACAGGCTCAATGGTGTC	21	59.8		
CL631	aCL631_F2	GCCTTGACGGTGAAAGATGAG	21	59.8	Induction of strobilation	(33)
	aCL631_R2	ACCTCGTCCTCATCCTTTTCG	21	59.8		
DNMT	aDNMT_F2	CATGCACAGTTCATCGTCGAG	21	59.8	Role in methylation of DNA	(33)
	aDNMT_R2	CAATCGAACGCCACGTAATGC	21	59.8		
RxR	aRXR_Q_F	AGATGTACAGTGCCACGTTGG	21	59.8	Induction of strobilation	(33)
	aRXR_Q_R	CAAGTGTTGAGTGCTTGACAG	21	59.8		
BMP	qBMP_F	GCAATGTAGCCCACTGCTTG	20	59.4	Secreted morphogen for cell differentiation and division	(41)
	qBMP_R	ATGCTTGGCCTTGGTCGAT	19	56.7		
Wnt1	Wnt1_F	AGATTTCTGGCGAAAAGGC	20	57.3	Morphogen for localization of anterior-posterior axis of strobilation segments	(42)
	Wnt1_R	TGTTTGCTGCACCATTGAG	20	57.3		
Ptc	Ptc_for	AGTGACAGCTCAACGAGTGG	20	59.4	Receptor in Shh signal pathway	(42)
	Ptc_rev	ACTCCGAGCAATGCTAACCA	20	57.3		

Revision of the manuscript "Asexual reproduction of *Aurelia aurita* depends on the presence of a balanced microbiome at polyp stage" by Nadin Jensen, Nancy Weiland-Bräuer, Shindhuja Joel, Cynthia Chibani, and Ruth Schmitz-Streit (**Subject: Spectrum00262-23**)

We thank the three independent reviewers for the review process. We are pleased that the reviews were generally positive. We carefully addressed all questions in the point-by-point response below (reviewer-specific comments are highlighted in italics, compared to our answers in regular font) and revised the original manuscript according to the reviewers' recommendations. We have marked the changes in the manuscript with different colors for each reviewer (Reviewer 1, yellow; Reviewer 2, green; and Reviewer 3, turquoise).

Point-by-point responses:

Response to Reviewer #1

*In this article Jensen and colleagues explore the impact of the complex microbiome on strobilation of the jellyfish *Aurelia aurita*. They find that the presence of the native microbiome is required at the polyp stage for normal strobilation and ephyra release, and that introduction of the microbiome at subsequent timepoints is insufficient for metamorphosis. This paper is well thought-out with very nice figures. Understanding of host-microbe interactions in complex mutualisms is an important topic and this paper advances that understanding. I have some comments (see attached) that I think should be addressed, but consider them fairly minor.*

We thank the reviewer for the review of our manuscript. We are pleased that only minor comments were raised and are thankful for the impression that we submitted a "well thought-out paper with very nice figures". Nevertheless, we are happy to answer the raised questions. Changes in the manuscript were accordingly highlighted in yellow.

General Comments:

1) I'd briefly explain around line 142 that the inoculum is from filtered homogenate.

Answer: We thank the reviewer for this valuable comment. We added the information in line 171 (track change version).

2) I liked your experiment demonstrating that cell-cell contact is required. My question building on this is whether you think tissue colonization by the bacteria is required, or whether you think bacteria in the water would be sufficient?

Answer: We thank the reviewer for his/her interest in our experiment. Indeed we propose that tissue colonization is required. In the meantime, we conducted preliminary experiments focusing on required tissue colonization for regular progeny output. We, therefore, incubated one sterile and one native polyp together in 1 mL sterile ASW for 72 h without a separating membrane in close vicinity before inducing strobilation with the chemical inducer. However, strobilation and ephyrae release failed. We further incubated sterile polyps in native ASW. Here, polyps were crucially diminished, as

shown in a previous publication (Weiland-Bräuer et al., 2020, mBio). Consequently, we assume that tissue colonization of the polyp epithelium by key bacteria is required for the onset of strobilation. Those results have to be further verified and will be included in a different manuscript

3) For lines 302-303 I would note that Kerwin et al. 2019 in mBio did find that chloramphenicol negatively impacted development of the Hawaiian bobtail squid embryo, but that other antibiotics did not, and that paper was looking at microbiome functionality. Not asking you to cite it, but wanted to point it out. :) I was a little confused by your description of the antibiotics tested in lines 451-458 - which were tested singly, and did those not work sufficiently? Given the known toxicity of chloramphenicol that you cite, it may have been better not to include that. I think though that since your recolonized sterile polyps were fine, the antibiotic treatment wasn't a problem.

Answer: We thank the reviewer for her/his thoughts on antibiotic use. Indeed, we established an antibiotic mixture to generate sterile polyps by using different concentrations and mixtures of antibiotics. First, all antibiotics were singly tested; however, no single antibiotic could remove all bacteria, although cell numbers were reduced. Further, we tested different combinations of antibiotics, but again, 16S rRNA PCR revealed that some bacteria were unaffected by the mix. Finally, only the mixture of chloramphenicol, neomycin, ampicillin, streptomycin, rifampicin (each 50 mg/L), and 60 mg/L spectinomycin resulted in germ-free polyps. We are aware that antibiotics can have potential side effects on the host, as shown by Kerwin et al. 2019 for chloramphenicol. However, other antibiotics did not affect the embryo, as pointed out, and chloramphenicol did not affect adult squids. We now included the respective reference in the revised manuscript (line 351). All used antibiotics (single and combination) showed the same morphological deformations of *A. aurita* polyps independently of their target sites. Besides, recolonization led to a normal phenotype. We thus assume that neither chloramphenicol nor other antibiotics affect the polyp and conclude that a manipulated microbiota causes the observed defects.

4) The NCBI BioProject ID did not appear when searched - assume this will be public by publication.

Answer: We thank the reviewer for this important comment. The reviewer is correct; we allow publicity of the dataset after the publication of the manuscript.

5) Did you do a glycerol control since your inoculum were frozen with glycerol? I'm assuming not, in which case you should note that somewhere and the reasoning behind that decision.

Answer: We thank the reviewer for bringing up this control. We proved the viability of cells after several freeze-thaw cycles without a significant loss of cell numbers due to the use of glycerol as a cryoprotectant. A glycerol-only control was conducted before the experimental start. A glycerol stock was prepared like the inoculum (10 % glycerol). Using the same concentrations as in the recolonization experiment (10 µl in 1 ml ASW), we incubated native and sterile polyps for 1 d to 7 d in the presence of glycerol. Polyp phenotypes and survival rates were not affected; thus, glycerol was rated neglectable. In the revised version, we have included this information in the Material&Methods part.

Figure Comments:

1) In Figure 1 I found it unclear what was happening in the sterile pathway of A - what do the 2nd and 3rd grey images represent? A description in the legend or in your text (around line 133) would be helpful (I see you get to this later on pg 9, but I think a brief description early on would still be helpful). The NC and SC abbreviations should be defined in the legend. Does the R1 ephyra not have a question mark because that was studied in your last paper? I was unclear why the inducer was shown outside of the box for R1 and why that was different with it being inside the box in R2/R3. Also why is it R1 and not R1i if the inducer was used? Should it say inducer instead of inductor in the figure?

Answer: We apologize for not clearly phrasing the figure legend and being unclear about the Figure itself. For clarification: The second and third grey images are the early and late strobila; the inducer for the native polyp is applied after recolonization leading to a normal ephyra output. Indeed, the R1 ephyra has no question mark, as we have studied this aspect in the last paper (Weiland-Bräuer et al., 2020; mBio). The inducer inside the box should indicate that when the bacterial mix was added for recolonization, the inducer was again added to have inducer and recolonizing bacteria always at the same time present (continuous inducer presence). We now revised the figure legend in accordance with the reviewer's comment and added a short description in the manuscript (lines 161-163).

The revised figure legend now states accordingly: "Figure 1. Life cycle of *Aurelia aurita* and study design for recolonization at various developmental stages. (A) Summary of the life cycle of the moon jellyfish *Aurelia aurita*. Sexual reproduction of the matured medusa results in the release of planula larvae that develop into benthic polyps. Asexual reproduction includes the transition into early and then late strobila, forming bodily segments to release precursor medusa in the form of ephyrae. This offspring generation is impaired in the absence of a microbiota. Sterile animals showed malformations characterized by a pale color, slim body shape, and lack of tentacles represented in a grey color of early and late strobila (11). (B) Experimental design to identify permissible time point(s) of recolonization. Native polyps are induced with the chemical inducer for the onset of strobilation (native conditions; NC), which was removed by washing after 72 h. Similarly, sterile polyps were induced, resulting in impaired strobilation (sterile conditions; SC). Sterile polyps were recolonized prior to induction (R1). Developed sterile strobila stages (light grey images) were recolonized after washing off the inducer (R2, R3); in a second approach, the inducer was again added when recolonization was performed (constant presence of inducer) (R2i, R3i)."

2) I really liked your color scheme for Figure 2 and the way you organized the taxa. For the bolded genera it would be good to define what your threshold was for "relative high abundance". The scroll over with mouse click functionality wasn't working for me but I assume that will be checked in the proof stage. I was surprised to see that the alpha diversity of the recolonized polyps was so much higher. This should be discussed more - do you have a hypothesis? It doesn't seem like you should be able to strongly increase in richness between what's being inoculated and what's being colonized - unless you suspect an increase in richness (which at least to me doesn't seem super apparent). You later call the diversity comparable (lines 327-329) which seems at odds with your data. Visually comparing the inoculum and the recolonized polyps I wondered if any taxa were missing - might be helpful to underline or otherwise indicate taxa present in the inoculum but missing in recolonized, or alternatively present in native but missing in recolonized.

Answer: We thank the reviewer for the comments to improve the Figure. Particularly, a scroll-over function would be achievable and implemented in the published version. The bold font, highlighting specific genera, resulted from notably changed genera comparing treatments. However, the taxa are already sorted by decreasing abundance. Since the message was misleading, we removed the bold font to highlight those genera. The threshold for showing taxa is over 1 % relative abundance; this information is now included in the figure legend. Taxa below this threshold are grouped as "others". Within this group "others", genera present in native polyps but missing in recolonized polyps are hidden. We changed the Figure and legend accordingly.

We were also surprised by the alpha diversity within recolonized polyps and have the following hypothesis. The diversity is higher in recolonized polyps than in native ones but with less variation. We depict only a snapshot of the microbial composition after 48 h of recolonization and assume that the microbiota adapts with time (> 48 h), resulting in higher variation due to biological replicates. In total, 160 more ASVs were detected in recolonized polyps than in native ones, explaining the higher alpha diversity. However, none of those ASVs represent taxa not present in native polyp communities. Moreover, 122 ASVs were present in native but not in recolonized polyps among those representatives of Alpha- and Gamma-Proteobacteria, Bacilli, Bacteroida, and others. Those ASVs are suggested as irrelevant to the reproduction success. The higher relative abundance in recolonized polyps might be explained by a high number of ASVs that are outcompeted over time. In a future study, we plan to analyze the microbial community of recolonized polyps at different time points (long-term experiment).

3) Figure 3 is really nicely done - great job!

Answer: We thank the reviewer for taking our efforts into account.

4) For part A of Figure 4, are you inducing to get ephyra in the SC, R2, R3 conditions? Or do you still get some ephyra, just a smaller percentage? The percentages in the ephyra boxes should be mentioned in the legend. For the late strobila, where is the color coming from in NC and R1? It makes it a little hard to compare to the SC and R2 since those appear more as silhouettes, but maybe that's just what they look like? (Sorry I'm not familiar with what they should look like in this species.)

Answer: We thank the reviewer for raising those questions. All polyps, native and sterile, were induced for the onset of strobilation. Animals were recolonized at different time points (polyp, early and late strobila). All animals formed strobilae and released ephyrae; however, indicated treatments showed malformed phenotypes and consequently malformed and significantly less formed ephyrae. Exclusively, native and recolonized polyps showed regular strobilation and ephyrae release with the expected numbers. The respective percentages of released ephyrae of the different later stages are depicted in the box compared to the native polyp (which is set to 100 %), now mentioned in the figure legend of the revised version. We agree with the reviewer that different treatments are hard to compare based on their appearance; however, the brownish color of prolonged polyps and strobilae is a sign of regular strobilation. In 1980, a study by van den Branden described that the brown-red color of strobilae and ephyrae is formed in the animal and is melanin accompanied by a smaller fraction of a protein-bound brown pigment. These pigments appear as soon as strobilation is started and disappear after the liberation of ephyrae from the strobila. The information was added in the revised manuscript.

5) Figure 5 is really nice - great presentation of complex data! How do you explain the

difference in expression between the native and R1 at the polyp stage? The Figure seems to contrast with your results at lines 253-254?

Answer: We thank the reviewer for the valuable comment. Indeed, a difference between native polyps and recolonized ones was detected for CL355. The differences between CL112, CL390, and CL631 were non-significant, although the color code in part A is a bit misleading. That the differences are not significant is more visible in part B of the Figure, where the expression pattern over time for one gene is exemplarily shown – indicating no difference compared to the native polyp. Since the function of the CL genes has not been elucidated, we can only speculate that the slightly different community patterns of native and recolonized polyps cause the difference in expression patterns of CL355. We assume that the CL genes are differentially expressed during strobilation and might have different functions and importance throughout the process, revealing a variance in the effects of the treatments. In lines 295 ff., we describe the transcription profile of sterile polyps, which contrasts with native and recolonized ones. Similarly, recolonization after strobilation onset impaired gene expression of CL genes.

6) Figure 6 refers to A/B/C but these labels aren't present in the Figure itself. For B in the legend you don't mention the inducer but the Figure shows it - I'd reword to make the legend clear. You need to define the scale bars in the legend.

Answer: We apologize for the incomplete Figure. We added the labels, the symbol for the inducer, and the scale bars. The legend was completed with the scale bar definitions.

Minor Comments:

- 1) In lines 26, 35, 46, 77, 238, 324 would it be better to say polyp-to-medusa?*
- 2) In line 28 I think it should be accompanied by instead of accompanied with.*
- 3) In line 65 I'm not sure whether the sentence starting Several hosts is necessary. I'm unclear as to what you mean here (unless it really is just that many different systems can be models?) so I'd either rephrase or drop it all together.*
- 4) In line 95 should be The expression, not These expression.*
- 5) In line 97 should be "and is expressed" not "and be expressed".*
- 6) In line 265 should it be "path" instead of "faith"? Or maybe "fate"?*
- 7) Sentence starting The abundance in lines 331-334 needs to be revised for clarity.*
- 8) Sentence starting Examples of in lines 341-342 is quite vague and could be revised to briefly provide the actual examples.*
- 9) I don't think oyster should be italicized in line 369 - might actually be better to provide the specific species.*
- 10) Line 369 has a typo - should be whether, not wether.*
- 11) Line 481 - revise sentence*

Answer: We thank the reviewer for the comments. All raised minor comments were accordingly changed in the revised version of the manuscript.

Reviewer #2

The research of host-microbiota interactions is very up-to-date and in particular this study as it brings new insights in novel model metaorganism, i.e., jellyfish, which is unlike other systems, e.g.,

corals, hydrozoans, etc, less studied metazoan host-microbiota system. Being ecologist my major concern, as always with model organisms that have been capture/kept in the lab for a long period of time, is the actual accuracy or relevance of results for ecology of this organisms. In my modest opinion these systems, like the case here, when organisms have been kept in the lab for over 10 years, are far away from their natural representatives. Else, study is very well designed, conducted and analysis is very appropriate and paper well structured (however a bit long and repetitive in some places and hence could be more condensed and streamlined). Hence, I suggest this paper can be accepted with minor revision. Please see my specific comments below.

Answer: We thank the reviewer for reviewing our manuscript and the ecological perspective brought in. We agree with the reviewer that one should be careful with conclusions from studies with model organisms kept in the lab. We carefully went through our manuscript and toned down conclusions on ecological relevance. Nevertheless, we believe that model studies are essential to gain first insights into aspects of physiology, development, and ecology. We are pleased to answer the raised minor comments below, addressed in the revised manuscript accordingly (see marked passages in green).

Some minor comments:

Abstract: Line 23: Yet, the timing and molecular consequences of the microbiome during the 24 strobilation process had not been investigated. Is strangely formulated – the timing and molecular consequences of the microbiome- please re-write.

Answer: We thank the reviewer for the comment. The abstract was extensively changed due to addressing the comments of Reviewer 3, including the mentioned phrase.

Introduction: Line 86-87: Several proteins, including transcriptional regulators have been shown to be involved in the induction of strobilation. Reference missing.

Answer: We apologize for the missing reference. We added it in the revised version (line 108).

Line 97-98: These expression of these four genes is strongly upregulated in polyps entering the strobilation phase. CL112 97 encodes a secreted protein containing an epidermal growth factor-like domain and be expressed in endodermal cells of each segment of the early strobili. Reference missing

Answer: We apologize for the missing reference that we added in the revised version (line 125).

Results/Methods:

General comment: can you please justify use of V1-V2 region? Some microbiota could be missed in this case. Did you try other primer sets and compare?

Answer: We thank the reviewer for the valuable comment. We are aware of the choice of different primer sets for microbial community analysis by amplicon sequencing of 16S rDNA. However, we are confident, we do not miss any taxa but might resolve their taxonomical classification differently (see below Rausch et al. 2019). When performing a taxonomic classification, we used the V1-V2 region based on the known valid proxies for full-length sequences (Werner et al., Nature, 2012). We decided on the V1-V2 primer set in previous studies and stayed with this decision for internal comparison. Moreover, a ring trial with several model organisms comparing all the different primer

pairs and amplification protocols revealed for *A. aurita* that V1-V2 primers performed better than V3-V4 primers (Rausch et al., 2019 Microbiome).

Methods:

Line 406: For isolation of A. aurita-associated microbiota, 10 polyps, 10 early strobilae (5 days 407 post-induction), and 10 late strobilae (9 days post-induction) were transferred into 500 µl sterile ASW and washed three times to remove transient bacteria. The animals were collectively homogenized with a motorized pestle (KONTES, DWK Life Science, Wertheim, Germany) and filtered through a 3.1 µm filter (Lab Logistic group, Meckenheim, Germany) to remove eukaryotic cells. This is not clear to me. Did you combine/pool microbiota from all this different 'hosts' to obtain inoculum or you kept it separately for each 'host'? Depending on what exactly you did in my opinion affects the outcome substantially.

Answer: We thank the reviewer for asking about details on microbiota generation. We used a pool of 10 carefully washed animals for each life stage, not mixing the different stages. Those animals from one developmental stage were homogenized, the homogenate filtered to remove eukaryotic cells, and the microbial cell numbers were counted to set the concentration. Life stage-specific generated microbiota stocks were stored and used separately. Based on the life stage-specific microbial community patterns, we aimed to exclude life stage-specific community effects for the recolonization experiment, including R2/R3, shown in Figure 4C. We now rephrased the sentence in lines 485 ff. for a better understanding in the revised version.

Line 412: Aliquots (105 cells/ml) of the filtrate were immediately frozen in liquid nitrogen and stored at - 80 °C in the presence of 10 % glycerol. These frozen stocks served as inoculum for all recolonization experiments presented here. I wonder how many of cells lost their viability via this process – maybe also specific taxa...did you maybe check the effect on microbial community?

Answer: We thank the reviewer for this valuable comment. We checked the effect of several freeze/thaw cycles on the viability of cells in initial experiments by counting the cell numbers and colony-forming units. No significant loss of viability was detected even after three cycles. The community analysis of the inoculum (see Fig. 2) represents the snapshot of the inoculum after thawing before the use within recolonization experiments. Here, 85 ASVs present in the community of native polyps were not detected in the inoculum, suggesting that they got lost during preparation or lost viability after thawing (non-cultivable strains missed in cfu counts). Nevertheless, those microbes probably play a minor role in the considered asexual reproduction of *A. aurita* since the recolonized polyps reproduced like the native polyps. The affected taxa are grouped within "others".

Line 482: Recolonization of sterile polyps, early and late strobila was conducted by addition of 10 µl of the thawed inoculum to ASW (105 cells/ml and per animal, see above), with fresh ASW containing inoculum applied the next day, so that the animals were exposed to the inoculum for 48 h. A control of inoculum was incubated for 48 h in ASW only. At this point I started to think about viable/dead cells within inoculum and then I realized that you used SYTo0 to count – which allows you to distinguish live/dead cells, however maybe not all readers know that so would be nice to specify this somewhere. Else makes you wonder how much viability of cells was affected by your freezing/thawing of inoculum, hence also affecting your community composition,

Answer: We thank the reviewer for the comment and refer to our previous answer. The revised manuscript includes a short note on Syto9 as a live cell dye (see lines 496 ff.). Moreover, live/dead staining was exclusively conducted with the microbiota of native polyps and the generated

microbiota (inoculum) but not with recolonized polyps. Consequently, we cannot compare cell numbers and relative abundance trends for the recolonized polyps, but we will implement such experiments in future studies.

Results: Line 149: what about alpha diversity of whole community?

Answer: We thank the reviewer for the comment. All analyses refer to whole communities of the specified samples. Alpha diversity is plotted in Fig. 2B for the microbiome of native polyps, the generated microbiota of polyps used as inoculum in recolonization experiments, the recolonized polyp, and the inoculum incubated in ASW. Interestingly, the alpha diversity is higher in the recolonized polyp. We elaborated and suggested potential reasons for this question above (reviewer 1, comment to Figure 2).

Line 152 (and other cases): when you talk about relative abundance one needs to be aware that of course this depends on total cell abundance, e.g., one taxa can increase in relative abundance, but overall cell abundance (cells/mL) is lower and this is important!, also an increase/decrease can be due to decrease/increase of other taxa...it can give a wrong impression! Since I understood you used SYTO9 to distinguish between live and dead and made an effort to count cells on polyps etc., why not making an effort to take this actual cell numbers into account

Answer: We thank the reviewer for the idea of accounting for the overall living cell abundance for the detected cell numbers. In consultation with bioinformaticians, we abstain from this recommendation. Since relative species abundance refers to how common or rare a species is relative to other species in a given location or community, the percentages can give insights into trends. However, they can not be calculated with detected cell numbers (due to the bioinformatic determination of species abundance). Theoretically calculating with living cell numbers might be misleading within a complex microbiota, as shown for *A. aurita*. Future studies should implement the detection of relevant key taxa by fluorescence in situ hybridization using taxon-specific probes. This would enable monitoring cell counts and localization in space and time.

Line 156-157: sentence not clear, please re-write: Overall, the microbiota of recolonized polyps more strongly resembled that of native polyps than bacteria maintained in artificial seawater (ASW) in absence of animals.

Answer: We apologize for the misunderstanding. We rephrased the sentence in the revised manuscript (lines 183 ff.).

Line 216-216: The individual administration of antibiotics also reduced the native bacterial community monitored by a decrease of 30 - 80 % in CFU/ml. This is confusing to me: when you talk about CFU/mL I assume only cultivable part of community was assessed? But to my understanding this was not the case. Please explain.

Answer: We thank the reviewer for the valuable comment. For all (single and combination) antibiotic treatments and community reduction determination, we proved the potential sterility of animals by colony-counting the cultivable part of the microbiota in the first step. Secondly, non-sterile samples were not further analyzed. However, exclusively those samples without any detectable CFU (assumed to be sterile) were used for 16S rRNA amplicon sequencing. Sterility was generally verified by amplification failure.

Line 270-272: For this, native polyps with their diverse, healthy microbiome were incubated together with sterile polyps, but separated by a 0.2 µm filter that allowed the transfer of metabolites, but not

of bacteria. If we are picky we would take into account the fact that some really small bacteria can escape 0.2 μm filtration!

Answer: We agree with the reviewer that very small microbes can escape 0.22 μm filtration. However, the potentially remaining microbes could not restore the impaired fitness effect of sterile polyps, indicating no crucial effect on the strobilation process.

Discussion:

Line 287: Our results not only illustrate how important a native microbiome is for the asexual reproduction of A. aurita, but also demonstrated that the generation of offspring is only possible when the microbiome is present before the animal enters the strobilation process. Just a general thought: I cannot image where and when in natural scenario microbiome would not be present before animal enters strobilation process, so not really sure, from ecological perspective, why this I even relevant. Maybe discussion on this aspect could be elaborate.

Answer: We thank the reviewer for the ecological point of view. We did not intend to discuss the importance and presence of microbes in general or even the occurrence of sterile surfaces in nature. We wanted to express that microbes, specifically key taxa and their related functions, are of enormous importance for the development of *A. aurita* at the specified time point (development stage). The only way to analyze that in experiments is to remove the microbes and elucidate the effects on asexual reproduction artificially. The presence of the specific polyp microbiome is crucial before the onset of strobilation for regular progeny output. We have therefore rephrased the sentence accordingly to avoid misunderstandings: the bacterial impact is required for normal offspring generation before the animal enters strobilation.

The bacterial impact is required before entering the strobilation process.

Line 332-334: The abundance of Vibrio and Colwellia is increasing in recolonized polyps, whereas ASW shifted the bacterial population to a significantly structure accompanied with a lower α-diversity. This is a very strange sentence, I don't understand what is meant here. Please correct.

Answer: We thank the reviewer for the valuable comment. The sentence combines two very different pieces of information, indicating a causality, which is not the case. We now changed the sentence accordingly (lines 393 ff.).

Line 336-337: During a reassembly of the microbiota on the host's mucus, interactions between host and bacteria as well as amongst bacteria ensure a specific microbial community pattern of Aurelia. How can you possibly know this? Did you test other jellyfish species and compare species-specificity of associated microbiota?

Answer: The point is well taken, and we apologize for not being precise enough here. A previous study showed that a substantial proportion of the *A. aurita* microbiome is attached to the mucus (Weiland-Bräuer et al. 2015). The microbes are assumed to be provided with nutrients, live in close vicinity, and can thus interact with each other and the host. One kind of bacterial interaction to maintain a specific microbiota might be quorum sensing/quenching, which we have published on (Weiland-Bräuer et al., 2020). We further know and reported on other jelly microbiomes. The microbiomes were jelly-specific. However, we did not prove species specificity by swap experiments (Jaspers et al., 2020). In addition, we hypothesize host attraction and defense mechanisms for establishing and maintaining the specific microbiota, again published in a previous paper (Weiland-Bräuer et al., 2019). In the present study, e.g., the community comparison of native polyps and the

inoculum in ASW (with and without the host) pointed to such a host impact. Although we have several hints on the impact of metaorganismal partners for establishing and maintaining the species-specific microbiota, we now toned down our statement in lines 399 ff.

Reviewer #3

Dear authors, this is an interesting study, however, the manuscript needs a lot of work; the good news - there is nothing that can't be fixed (see comments in the manuscript file). The original title is misleading, general concepts of the jellyfish lifecycle are either not well understood or not precisely worded, the abstract is confusing and lacks any description of methods, there is a lack of understanding which parts belong into which section of the paper, referencing is done in a sloppy way - it's not enough to reference a study, it also needs to be put into context. There are conclusions that are not supported by the results. There is an overuse of specific words, e.g. the word crucial appears about 15 times. While the results are worth publishing, the importance of the study is generally overstated and the wording is disproportionally strong and often exaggerated to a point where it does not reflect the findings. The shortcomings of the study need to be elaborated on (e.g. husbandry conditions). Many sentences are highly confusing, thoughts need to be ordered and sentences reworded. There are grammatical and style issues and I do recommend a proper revision of the English, if not by a native speaker at least by an online grammar and punctuation checker. I feel that the manuscript has been submitted either prematurely, or in a rush, which is kind of disrespectful to the reviewers, please consider this in your future submissions. For more details see the manuscript file.

We thank the reviewer for the intensive and valuable review of our manuscript. We apologize for the impression of submitting a premature manuscript, which is definitely not the case. A professional, scientific writer and native speaker supported the writing process. We further apologize for the "sloppy referencing". Nevertheless, we appreciate the time-consuming review of our manuscript and are thankful for helping to improve it. In many passages, we have accepted the reviewer's suggestions to change the text, improve the language and tune down some statements. However, we have also decided at some places to maintain and stick to our writing style if no comments on the content were affected. The changes in the manuscript are highlighted in turquoise. Specific questions and comments are answered in the following point-by-point response. Lastly, we have gone through the manuscript carefully, and, despite the minor comments of two independent reviewers, we have made substantial changes based on the comments of Reviewer 3. We hope the revised version of the manuscript is now appropriate for publishing.

Abstract:

Line 40: You need to add your methods here. What did you actually do?

Answer: We thank the reviewer for the comment. However, the journal author's recommendations do not demand a separate mention in the abstract of the methods used. In our opinion, and that of two reviewers, by purely naming the methods, the abstract provides sufficient information to understand the study. We added additional information that qRT-PCR determined the expression of key genes.

Importance:

Line 55: You are referring to strobilation that is chemically initiated, right? Does strobilation occur at all with an absent microbiome and without artificially inducing it?

Answer: We thank the reviewer for the valuable comment. Naturally, strobilation of native polyps is induced by lowering the temperature to 10 °C starting within 20-30 days. Sterile polyps do not strobilate by lowering the temperature. However, using the chemical inducer initiates the strobilation of native polyps within three days. Sterile polyp strobilation is delayed for a couple of days, and exclusively malformed strobilae are formed with a reduced size, colorless appearance, and reduced number of segments, and consequently, only around 15 % of all malformed strobilae release one malformed ephyra. The impression arises that the chemical inducer is more potent to induce strobilation than lowering the temperature, maybe due to the concentration and "forces" the polyps to induce strobilation, which fails through the absence of a microbiome.

Line 59: You are definitely overusing this word! A strong word loses its meaning if used in every other sentence. Try thesaurus for a synonym and replace.

Answer: We thank the reviewer for the comment. We changed the text accordingly. "Crucial" is now mentioned four times.

Line 61: Reduced or absent transcription levels for sterile polyps?

Answer: In the original version, the text mentioned reduced transcription levels for sterile polyps.

Line 63: Be precise on which asexual reproductive mechanism you are talking about.

Answer: We thank the reviewer for the recommendation to mention the asexual reproduction mechanism precisely. We changed "asexual reproduction" into "asexual reproduction by strobilation" throughout the revised manuscript.

Line 64: Define native polyps.

Answer: Native polyps are defined as polyps of sub-population Roscoff kept in 30 PSU artificial seawater in 2 L plastic tanks in the laboratory, which clonally reproduce in the lab. Native polyps harbor a diverse and complex microbiome that was not manipulated. We have defined "native polyps" in the Material&Methods section in the revised manuscript (line 479).

Introduction:

Line 81: Why this one in particular? Please argue your point.

Answer: We rewrote the sentence: "The research on invertebrate-microbiota interactions, in particular, broadens the concept of metaorganism since it enables to disentangle fundamental mechanisms of host-microbe interactions as well as their underlying regulatory principles" to argue the point.

Line 138: Which are these multiple developmental stages? How many have you assessed? To my knowledge only one – strobilation, early and late. Be precise.

Answer: We tuned down to "several developmental genes" and added in brackets "assessed developmental stages polyp, early, and late strobila".

Line 148: Like which ones?

Answer: We added examples of phenotypic effects accordingly.

Line 150: How? Give a short and general introduction to the methods used.

Answer: We now named the qRT PCR method in the revised manuscript but abstained from explaining a well-known method.

Results:

Complete first paragraph: Introduction, methods, discussion: Remove this whole paragraph, moving the relevant parts to the respective sections.

Answer: We thank the reviewer for the comment. However, we categorized this comment as a criticism of our writing style and decided not to move those parts to different sections. In our view, these passages summarize complex data and context for the reader, and provide a better understanding of each paragraph and the manuscript as a whole.

Line 164: "Recolonization of sterile polyps with a native microbiota demonstrates a strong host impact on the microbial community structure" I don't understand what this is supposed to mean. Please clarify.

Answer: We thank the reviewer for the valuable comment. We want to highlight our evidence that the microbiota shows a massively changed composition when it is associated with the host compared to incubation in artificial seawater. Consequently, the host impacts the community structure by offering a niche with nutrient supply, presenting a surface where microbes live in close vicinity for interactions among those. Moreover, the host might possess attraction and defense mechanisms to influence the microbial community structure.

Line 169: Define truly native polyps., Line 171: Are these the ones you call truly native?

Answer: Please see our answer above (Line 64).

Line 189: This is well established knowledge, not a finding of yours. You could say "which corroborates with previous studies ..." and add a reference to it, or remove it altogether.

Answer: We thank the reviewer for the recommendation. We deleted the part of the sentence in the revised manuscript.

Line 192: Do not assume! Not part of the results section, but discussion.

Answer: We changed the word "assume" to "predict" but abstained from moving the section due to the explained type of our writing style (see above).

Line 197/198: Life cycle decisions??! What is that supposed to mean? The jellyfish makes life cycle decisions??! Please explain and reword.

Answer: We thank the reviewer for this valuable comment. We changed the wording in the following way: "In order to uncover whether and how the microbial presence influences the transition of life stages within the life cycle of *A. aurita* on the molecular level, ...".

Line 199: By who?? Reference.

Answer: We apologize for the missing references, which were added in the revised version.

Lines 200-203: How did you do this? Unclear. Also, awkward wording. Please re-write., So the animals are abnormal and you are building your argument of absent gene expression on assuming when "normal developmental stages" would occur?? That is a flawed argument.

Answer: We thank the reviewer for the comment. The reviewer understood the experimental design correctly. We compared the development of sterile polyps and strobilae with the native counterparts to set a fixed time point of analysis. Since we followed the development by morphological observations for further four weeks, we were sure not to miss the progress of strobilation under sterile conditions. Sterile polyps formed malformed strobilae, which phenotypically remained in the same status with no progress of strobilation. Transcription levels were consequently analyzed at the time points selected from native development. Transcription levels were significantly reduced under sterile conditions in all monitored developmental stages (polyp, early strobila, late strobila, and ephyra) of the life cycle. With this, we certainly observed a potential developmental delay under sterile conditions over 12 days.

The text is now changed in: "Since the life cycle of sterile animals is disturbed, the latter were analyzed at the time points where the normal developmental life stages would appear, although phenotypically the sterile animals were not at that stage and showed an abnormal morphology."

Line 203: Before or after strobilation?

Answer: We apologize for the missing information, which was added in the revised manuscript.

Line 209/210: Not a surprise. The animal in absence of its microbiome is likely in survival mode, unable to spare the energy needed for reproduction.

Answer: We agree with the reviewer that in the absence of the microbiome, the animal enters a "survival mode"; however, the present study is the first to show the consequences of microbiome loss on the molecular level, particularly by differential expression of strobilation genes.

Line 212: The analysis was extended? What does that mean?

Answer: We apologize for the miswording. We performed the transcription analysis by qRT-PCR also with general developmental genes. We changed the text accordingly.

Line 216: What does this mean? (fold changes)

Answer: We thank the reviewer for this question. Quantitative Reverse Transcriptase (qRT)-PCR measures transcript levels by detecting fluorescence signals. As measures, threshold cycle (Ct) values are revealed. The higher the Ct value, the less transcribed is the specific gene. The first step calculates a ΔCt value by comparing the specific gene's Ct values and a constitutively expressed housekeeping gene. The second step calculates a $\Delta\Delta Ct$ value for the desired comparison, e.g., native versus sterile conditions. The fold change is finally the expression ratio. Positive fold changes mean the gene is upregulated, while negative fold changes mean, it is downregulated (Livak and Schmittgen 2001).

Lines 226-230: Do we know exactly how many genes are involved in strobilation? Does the absence of 4 genes justify the description "severely impacted"? Maybe these are redundant ones ... I really think you need to tone this down – severely ...

Answer: We apologize for over toning the statement. We changed the manuscript following your recommendation and mention the number of genes.

Line 231: "Timing of the microbial contribution is crucial for the life cycle decisions of A. aurita: As mentioned in an earlier occasion.

Answer: We reworded "life cycle decision" into a more passive-sounding "transition of life stages" process.

Line 233: Which are the various stages? Name them.

Answer: The missing information was added in the revised version.

Line 236: Compared to what?? What did you compare?

Answer: We added the missing information about the comparison to native husbandry conditions.

Line 237: What does this mean? Native polyps? Wouldn't that be NP? Also, if this is the abbreviation, introduce it the first time the term is used in the manuscript.

Answer: We apologize for not introducing the abbreviation NC, which is now introduced in the revised version. NC describes the native conditions.

Line 250/251: This is unclear. The individual administration of antibiotics to what?

Answer: We apologize for the imprecise wording. We rephrased it in the following way: "single antibiotics were added to native polyps to generate sterile ones".

Line 253: When were they recolonized?

Answer: We apologize for not mentioning the recolonization time point we made up in the revised version. Sterile polyps were generated by incubation in an antibiotic mixture. Antibiotics were washed out, and polyps were subsequently recolonized with the generated microbiota.

Lines 270-274: Interpretations of results belong in the discussion! Remove all of these from the results section.

Answer: We abstain from moving this part to the discussion. As already explained, we will stick to our writing style.

Lines 277-286: Methods/ Interpretations of results belong in the discussion! Remove all of these from the results section.

Answer: We abstain from moving this part. As already explained, we will stick to our writing style.

Lines 287-289: This belongs in the methods section.

Answer: We abstain from moving this part. As already explained, we will stick to our writing style. Moreover, this section provides the necessary information to understand Figure 5.

Lines 292-294: This is the results section; describe each of your results in detail, not just one example.

Answer: We thank the reviewer for the comment. The reviewer is right that we focused on the detailed description of the differential expression of CL390. However, this example explains the general trend observed for all genes mentioned in the manuscript. We now mention this more pronounced (general trend), but beyond, we do not see the need to explain each gene transcription in detail since it is depicted in the Figure.

Line 297: What does this mean?

AND

Line 311: You have to reword this. It's not the monitoring that resulted in malformation.

Answer: We reworded the sentences to avoid misunderstanding.

Line 314: You cannot just conclude this!!! Only if you do an experiment that excludes the separation by 0.2 filter and another one that allows sterile polyp/native polyp direct contact you can comment on this. You could add in your discussion that these experiments have not been done but could potentially answer some interesting questions., Firstly, this has no place in the results section but belongs into the discussion, secondly only mention this if you have done the experiment. Otherwise, it's pure speculation without any proof.

Answer: We apologize for the misunderstanding. In lines 308 ff., we describe a conducted experiment. Native and sterile polyps were separated by a membrane, not allowing for cell but metabolite transfer. Since sterile polyps did not restore the impaired strobilation phenotype, we assume that direct contact with bacteria (e.g., a T6SS, or vesicles) is essential for regular strobilation. We agree with the reviewer that these are just preliminary results; however, those results point to a specific mechanistic interaction that has to be validated in future studies. Thus, the discussion section further includes our ideas on this potential interaction.

Lines 319-324: This belongs in the discussion part.

Answer: At this point, we would stay true to our writing style and make no changes.

Discussion:

Line 328: Again, you have only assessed strobilation which is ONLY ONE possible way of asexually reproduction in A. aurita. Others are budding, podocyst formation and many more. See Vagelli (2007) New observations on the asexual reproduction of Aurelia aurita (Cnidaria, Scyphozoa) with comments on its life cycle and adaptive significance.

Answer: We apologize for not being precise. We changed the unprecise wording of asexual reproduction to asexual reproduction by strobilation throughout the manuscript.

Line 347-352: Neglect means - fail to care for properly. Don't you agree that this is inappropriately strong wording? Those scientists were simply investigating another topic, not neglecting anything. Also, their work was done at a time long before the holobiont concept and /or microbiomes were a thing. You should simply say that it's the first time the microbiome has been considered in this context.

Answer: We apologize for the incorrect wording and agree with the reviewer. We changed the sentence accordingly.

Line 355: Only the ones that are visible to you.

Answer: We apologize for the misleading wording. We toned the sentence down.

Line 361: You are only mentioning one here. If you want to say "for other invertebrates", you need to mention more examples and references. Also, I would prefer to see references from organisms closer related to jellyfish, or at least marine invertebrates. A lot of research has been done on corals; can you find an example related to coral?

Answer: We agree with the reviewer on including more studies and references. We included an example of corals to support our statement (lines 369-371).

Line 372: What's the relevance of this quote? What are these proteins important for?

Answer: We thank the reviewer for the comment. The reference to the expression of mucus regulatory proteins in mice is an example of differential gene expression in native and sterile animals. Furthermore, mucus components are crucial in the interplay between the microbiota and the host in early-branching metazoans since mucus components are essential in developing the epithelial barrier as part of the innate immune defense.

Line 379: Lee at al. Is not about A. aurita but Chrysaora plocamia. Either remove or correct to "Some jellyfish polyps harbor ..." or "Polyps from some jellyfish species, such as A. aurita and Chrysaora plocamia harbor ..."

Answer: We thank the reviewer for this comment. We added the species names in the revised version, to be precise.

Line 388: Reference

Answer: We apologize for the missing reference, which was added to the manuscript.

Lines 390-394: This belongs in the results section. Irrelevant here, remove.

Answer: We categorize this part again as writing style and abstain from moving this part.

Line 395-399: This is a very confusing sentence, please reword and add a reference.

Answer: We apologize for the confusion. We revised the sentence and added a reference.

Line 402: Whose neighbours?? Other microbes, other Nematostella, other species ...? Please clarify.

Answer: We apologize for not being precise. We added the information that bacterial neighbors are meant.

Line 403/404: Which ones? Rather than just leaving a reference, put it into context, mentioning which interactions and/or hosts you are talking about.

Answer: We apologize for not mentioning the interactions and partners involved. We rewrote the sentence with more details.

Line 413: If you say "interestingly", explain why it is interesting, relevant and/or worth mentioning.

Answer: We stated why this aspect is interesting in the revised version. The text passage sounds: "In our experiments, certain bacteria were missing in the microbiota of recolonized polyps compared to native polyps, indicating redundant microbiome members, or at least members not required for normal strobilation. Interestingly, recolonization increased the abundance of *Vibrio* compared to native polyps assuming a potential function of *Vibrio* for the strobilation (Fig. 2A)."

Line 415-420: So? In which context? Related to what? Vibrio are almost everywhere., Explain what this reference is about., Which ones?, Not sure what this sentence is trying to say. Please clarify.

Answer: We thank the reviewer for the comment. We decided to delete this paragraph in the revised version since the increased relative abundance of *Vibrio spp.* in the recolonized polyp might only present a snapshot, which has to be further analyzed in future studies.

Line 433: Explain which organism this is. There might be readers who don't know.

Answer: We apologize for missing the domain. We included the information in the revised manuscript.

Line 444: You cannot assume. You need to do the experiment to say this!! Also, how long did this experiment run? Was there even enough time for some transfer? To properly asses what you are trying to infer here, you need an experimental setup with a) native-sterile polyp contact (animals can touch each other), b) native -sterile polyps in the same recipient without touching each other, c) native-sterile polyps separated through a mesh. Assess on a realistic timeline; use metabolomics to determine metabolites in the ASW. Maybe you could do this experiment in your working group sometime in the future, would be nice to see the results.

Answer: We thank the reviewer again for the comment and refer to our answer for comment on line 314. Besides, we are planning to perform a metabolomics analysis. However, this is not performed yet and not the focus of this manuscript.

Line 447: Or the sterile polyp receives what is needed from the native polyp, unrelated to the microbiome, also a possibility ...

Answer: We thank the reviewer for this idea. In the meantime, we have preliminary results that the pure presence of a sterile polyp and the microbes in the ASW can not restore the impaired strobilation phenotype.

Line 451: You don't know that. Just because they co-occur doesn't mean that they (or only they) control the reproduction phenotype.

Answer: We agree with the reviewer and toned down the statement, precisely mentioning that the presence of bacteria plays one role in the onset of strobilation.

Line 452: You didn't assess interaction, but presence.

Answer: We apologize for not being precise. We changed the sentence accordingly.

Methods:

Line 473: I would like you to add a short paragraph on the possible effects of such husbandry conditions, mentioning that the findings of this study are purely related to artificial conditions and may not apply to what is happening in natural ecosystems.

Answer: We thank the reviewer for the valuable comment. In the discussion part, we added a short paragraph on the husbandry conditions and their relatedness to the natural ecosystem (lines 424 ff.).

Line 477: How big are the containers? Are the polyp individuals kept separately? Light-dark conditions? Aquarium system? Flow through or stagnant? Is the temperature always the same? Lots of information is missing here. It's not enough to just say "as described in"; give at least some basic information.

Answer: We apologize for the missing information. We give more details on the husbandry of polyps in the Materials&Methods part of the revised manuscript.

Line 532: Were these identified?

Answer: We thank the reviewer for the valuable comment. We did not analyze the cultivated bacteria remaining after antibiotic treatment using 16S rRNA because it was not the focus of our study.

Line 542: Was this procedure done for each polyp involved in your experiment, or just once in the process of establishing the sterilization-protocol? Please add this information.

Answer: We apologize for not stating the information. We randomly selected six animals each (sterile polyps, early strobilae, and late strobilae) for the 16S rRNA analysis to verify the sterility of animals. Whole animals were homogenized and used for DNA isolation prior to the PCR and thus were not used for morphological and molecular monitoring. Consequently, we could not check all replicates from the experiment for sterility and assumed all remaining replicates as sterile. Nevertheless, we not only proved sterility for establishing the protocol but also for the recolonization experiments.

Line 546: How?

Answer: We apologize for the missing information here; however, the information on the recolonization of animals is explained in detail in the following manuscript section.

Line 552: How do these observations differ from strobilation stages and times in sterile polyps? I thought that strobilation was inhibited without the microbiome, but now you say that sterile polyps undergo all strobilation stages and produce ephyrae? Please clarify and add the information.

Answer: We apologize for the misunderstanding. We refer to our answer for the comment on the importance (line 55).

Line 560: I don't think that too many conclusions can be drawn from a study where sterile polyps from a 15-year-o lab culture in ASW are artificially induced to strobilate in a tiny body of water and then re-inoculated with the culturable fraction, which is much lower than the actual non-culturable microbiota, of non-sterile polyps

Answer: We apologize for the misunderstanding. We want to clarify that the inoculum used for the recolonization of sterile animals was the generated microbiota of the respective life stage, including cultivable and non-cultivable bacteria. Concerning the conclusions drawn from artificial settings, we refer to our answer for comment to line 473.

Line 563: What does this mean? Prior to artificially inducing strobilation chemically? Please add this information

Answer: We apologize for the misunderstanding. Treatment R1 includes sterile polyps that were recolonized for 48 h with the generated microbiota and subsequently induced to strobilate using the chemical inducer.

Line 564: How did you determine these in sterile polyps?

Answer: We thank the reviewer for this valuable comment. We refer to the answer given for comment on line 200.

Line 566: Is this another treatment? If so, please say so, e.g., "... we assessed polyps with and without the constant presence of the inducer."

Answer: We thank the reviewer for the suggestion and revised the sentence accordingly.

Line 578: Why? You never mentioned this before? Please introduce in more detail, what you are talking about in this paragraph.

Answer: We thank the reviewer for the comment. Please see our answer to comment 314.

Line 587: So that's a total of 120 experimental units? How many individuals per experimental unit?

Answer: We thank the reviewer for the comment. We analyzed the expression with six animals for each developmental stage (polyp, early strobila, late strobila, and ephyra) ($\Sigma 24$). Furthermore, five treatments (NC, SC, R1, R2, and R3) were assessed, describing the different recolonization time points ($\Sigma 120$). A total of 120 individuals were analyzed, and nine genes were regarded.

Line 589: Assessed or identified/found? And if assessed, why those? Selection criteria?

Answer: We thank the reviewer for the comment. We selected ten genes (one housekeeping gene, four strobilation genes, and five developmental genes) based on the studies of Fuchs et al. and Wang et al., which showed the involvement and expression of those genes within the regulation of polyp-to-jellyfish transition in *A. aurita*. We assessed their transcription levels throughout strobilation after various recolonization scenarios.

June 6, 2023

Prof. Ruth A. Schmitz
Christian-Albrechts-Universität zu Kiel
Institute for General Microbiology
Am Botanischen Garten 1-9
Kiel 24118
Germany

Re: Spectrum00262-23R1 (The life cycle of *Aurelia aurita* depends on the presence of a microbiome in polyps prior to the onset of strobilation)

Dear Prof. Ruth A. Schmitz:

Thank you for your patience. Two of the original reviewers did provide feedback on the revised manuscript. Your manuscript has been accepted, and I am forwarding it to the ASM Journals Department for publication. You will be notified when your proofs are ready to be viewed.

Sincerely,

Kevin R. Theis
Editor, Microbiology Spectrum